# Time Series as Images: Vision Transformer for Irregularly Sampled Time Series

**Zekun Li, Shiyang Li, Xifeng Yan**
University of California, Santa Barbara
{zekunli, shiyangli, xyan}@cs.ucsb.edu

## Abstract

Irregularly sampled time series are increasingly prevalent, particularly in medical domains. While various specialized methods have been developed to handle these irregularities, effectively modeling their complex dynamics and pronounced sparsity remains a challenge. This paper introduces a novel perspective by converting irregularly sampled time series into line graph images, then utilizing powerful pre-trained vision transformers for time series classification in the same way as image classification. This method not only largely simplifies specialized algorithm designs but also presents the potential to serve as a universal framework for time series modeling. Remarkably, despite its simplicity, our approach outperforms state-of-the-art specialized algorithms on several popular healthcare and human activity datasets. Especially in the rigorous leave-sensors-out setting where a portion of variables is omitted during testing, our method exhibits strong robustness against varying degrees of missing observations, achieving an impressive improvement of 42.8% in absolute F1 score points over leading specialized baselines even with half the variables masked. Code and data are available at https://github.com/Leezekun/ViTST.

## 1   Introduction

Time series data are ubiquitous in a wide range of domains, including healthcare, finance, traffic, and climate science. With the advances in deep learning architectures such as LSTM [13], Temporal Convolutional Network (TCN) [18], and Transformer [38], numerous algorithms have been developed for time series modeling. However, these methods typically assume fully observed data at regular intervals and fixed-size numerical inputs. Consequently, these methods encounter difficulties when faced with irregularly sampled time series, which consist of a sequence of samples with irregular intervals between their observation times. To address this challenge, highly specialized models have been developed, which require substantial prior knowledge and efforts in model architecture selection and algorithm design [24, 20, 3, 16, 49, 34, 48].

In parallel, pre-trained transformer-based vision models, most notably vision transformers,[1] have emerged and demonstrated strong abilities in various vision tasks such as image classification and object detection, nearly approaching human-level performance. Motivated by the flexible and effective manner in which humans analyze complex numerical time series data through visualization, we raise the question: *Can these powerful pre-trained vision transformers capture temporal patterns in visualized time series data, similar to how humans do?*

To investigate this question, we propose a minimalist approach called **ViTST** (Vision Time Series Transformer), which involves transforming irregularly sampled multivariate time series into line graphs, organizing them into a standard RGB image format, and finetuning a pre-trained vision

---

[1]In this paper, we use the term "vision transformers" to denote a category of pre-trained transformer-based vision models, such as ViT [9], Swin Transformer [22], DeiT [36], etc.

37th Conference on Neural Information Processing Systems (NeurIPS 2023).

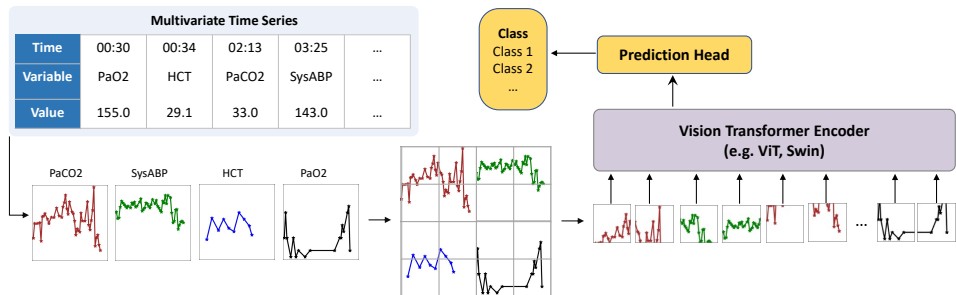

Figure 1: An illustration of our approach ViTST. The example is from a healthcare dataset P12 [12], which provides the irregularly sampled observations of 36 variables for patients (we only show 4 variables here for simplicity). Each column in the table is an observation of a variable, with the observed time and value. We plot separate line graphs for each variable and arrange them into a single image, which is then fed into the vision transformer for classification.

transformer for classification using the resulting image as input. An illustration of our method can be found in Figure 1.

Line graphs serve as an effective and efficient visualization technique for time series data, regardless of irregularity, structures, and scales. They can capture crucial patterns, such as the temporal dynamics represented within individual line graphs and interrelations between variables throughout separate graphs. Such a visualization technique benefits our approach because it is both simple and intuitively comprehensible to humans, enabling straightforward decisions for time series-to-image transformations. Leveraging vision models for time series modeling in this manner mirrors the concept of **prompt engineering**, wherein individuals can intuitively comprehend and craft prompts to potentially enhance the processing efficiency of language models.

We conduct a comprehensive investigation and validation of the proposed approach, ViTST, which has demonstrated its superior performance over state-of-the-art (SoTA) methods specifically designed for irregularly sampled time series. Specifically, ViTST exceeded prior SoTA by 2.2% and 0.7% in absolute AUROC points, and 1.3% and 2.9% in absolute AUPRC points for healthcare datasets P19 [29] and P12 [12], respectively. For the human activity dataset, PAM [28], we observed improvements of 7.3% in accuracy, 6.3% in precision, 6.2% in recall, and 6.7% in F1 score (absolute points) over existing SoTA methods. Our approach also exhibits strong robustness to missing observations, surpassing previous leading solutions by a notable 42.8% in absolute F1 score points under the leave-sensors-out scenario, where half of the variables in the test set are masked during testing. Furthermore, when tested on regular time series data including those with varying numbers of variables and extended sequence lengths, ViTST still achieves excellent results comparable to the specialized algorithms designed for regular time series modeling. This underscores the versatility of our approach, as traditional methods designed for regularly sampled time series often struggle with irregularly sampled data, and vice versa.

In summary, the contributions of this work are three-fold: (1) We propose a simple yet highly effective approach for multivariate irregularly sampled time series classification. Despite its simplicity, our approach achieves strong results against the highly specialized SoTA methods. (2) Our approach demonstrates excellent results on both irregular and regular time series data, showcasing its versatility and potential as a general-purpose framework for time series modeling. It offers a robust solution capable of handling diverse time series datasets with varying characteristics. (3) Our work demonstrates the successful transfer of knowledge from vision transformers pre-trained on natural images to synthetic visualized time series line graph images. We anticipate that this will facilitate the utilization of fast-evolving and well-studied computer vision techniques in the time series domains, such as better model architecture [23], data augmentation [32], interpretability [4], and self-supervised pre-training [15].

## 2    Related work

**Irregularly sampled time series.**  An irregularly sampled time series is a sequence of observations with varying time intervals between them. In a multivariate setting, different variables within the

same time series may not align. These characteristics present a significant challenge to standard time series modeling methods, which usually assume fully observed and regularly sampled data points. A common approach to handling irregular sampling is to convert continuous-time observations into fixed time intervals [24, 20]. To account for the dynamics between observations, several models have been proposed, such as GRU-D [3], which decays the hidden states based on gated recurrent units (GRU) [6]. Similarly, [46] proposed an approach based on multi-directional RNN, which can capture the inter- and intra-steam patterns. Besides the recurrent and differential equation-based model architectures, recent work has explored attention-based models. Transformer [38] is naturally able to handle arbitrary sequences of observations. ATTAIN [50] incorporates attention mechanism with LSTM to model time irregularity. SeFT [16] maps the irregular time series into a set of observations based on differentiable set functions and utilizes an attention mechanism for classification. mTAND [34] learns continuous-time embeddings coupled with a multi-time attention mechanism to deal with continuous-time inputs. UTDE [48] integrates embeddings from mTAND and classical imputed time series with learnable gates to tackle complex temporal patterns. Raindrop [49] models irregularly sampled time series as graphs and utilizes graph neural networks to model relationships between variables. While these methods are specialized for irregular time series, our work explores a simple and general vision transformer-based approach for irregularly sampled time series modeling.

**Numerical time series modeling with transformers.** Transformers have gained significant attention in time series modeling due to their exceptional ability to capture long-range dependencies in sequential data. A surge of transformer-based methods have been proposed and successfully applied to various time series modeling tasks, such as forecasting [19, 51, 41, 52], classification [47], and anomaly detection [45]. However, these methods are usually designed for regular time series settings, where multivariate numerical values at the same timestamp are viewed as a unit, and temporal interactions across different units are explicitly modeled. A recent work [25] suggests dividing each univariate time series into a sequence of sub-series and modeling their interactions independently. These methods all operate on numerical values and assume fully observed input, while our proposed method deals with time series data in the visual modality. By transforming the time series into visualized line graphs, we can effectively handle irregularly sampled time series and harness the strong visual representation learning abilities of pre-trained vision transformers.

**Imaging time series.** Previous studies have explored transforming time series data into different types of images, such as Gramian fields [39], recurring plots [14, 37], and Markov transition fields [40]. These approaches typically employ convolutional neural networks (CNNs) for classification tasks. However, they often require domain expertise to design specialized imaging techniques and are not universally applicable across domains. Another approach [35] involves utilizing convolutional autoencoders to complete images transformed from time series, specifically for forecasting purposes. Similarly, [31] utilized CNNs to encode images converted from time series and use a regressor for numerical forecasting. These approaches, however, still need plenty of specialized designs and modifications to adapt to time series modeling. Furthermore, they still lag behind the current leading numerical techniques. In contrast, our proposed method leverages the strong abilities of pre-trained vision transformer to achieve superior results by transforming the time series into line graph images, sidestepping the need of prior knowledge and specific modifications and designs.

## 3 Approach

As illustrated in Fig. 1, ViTST consists of two main steps: (1) transforming multivariate time series into a concatenated line graph image, and (2) utilizing a pre-trained vision transformer as an image classifier for the classification task. To begin with, we present some basic notations and problem formulation.

**Notation.** Let $\mathcal{D} = \{(\mathcal{S}_i, y_i) | i = 1, \cdots, N\}$ denote a time series dataset containing $N$ samples. Every data sample is associated with a label $y_i \in \{1, \cdots, C\}$, where $C$ is the number of classes. Each multivariate time series $\mathcal{S}_i$ consists of observations of $D$ variables at most (some might have no observations). The observations for each variable $d$ are given by a sequence of tuples with observed time and value $[(t_1^d, v_1^d), (t_2^d, v_2^d), \cdots, (t_{n_d}^d, v_{n_d}^d)]$, where $n_d$ is the number of observations for variable $d$. If the intervals between observation times $[t_1^d, t_2^d, \cdots, t_{n_d}^d]$ are different across variables or samples, $\mathcal{S}_i$ is an irregularly sampled time series. Otherwise, it is a regular time series.

**Problem formulation.** Given the dataset $\mathcal{D} = \{(\mathcal{S}_i, y_i)|i = 1, \cdots, N\}$ containing $N$ multivariate time series, we aim to predict the label $\hat{y}_i \in \{1, \cdots, C\}$ for each time series $\mathcal{S}_i$. There are mainly two components in our framework: (1) a function that transforms the time series $\mathcal{S}_i$ into an image $x_i$; (2) an image classifier that takes the line graph image $x_i$ as input and predicts the label $\hat{y}_i$.

## 3.1 Time Series to Image Transformation

**Time series line graph.** The line graph is a prevalent method for visualizing temporal data points. In this representation, each point signifies an observation marked by its time and value: the horizontal axis captures timestamps, and the vertical axis denotes values. Observations are connected with straight lines in chronological order, with any missing values interpolated seamlessly. This graphing approach allows for flexibility for users in plotting time series as images, intuitively suited for the processing efficiency of vision transformers. The practice mirrors **prompt engineering** when using language models, where users can understand and adjust natural language prompts to enhance the model performance.

In our practice, we use marker symbols "$*$" to indicate the observed data points in the line graph. Since the scales of different variables may vary significantly, we plot the observations of each variable in an individual line graph, as shown in Fig. 1. The scales of each line graph $g_{i,d}$ are kept the same across different time series $\mathcal{S}_i$. We employ distinct colors for each line graph for differentiation. Our initial experiments indicated that tick labels and other graphical components are superfluous, as an observation's position inherently signals its relative time and value magnitude. We investigated the influences of different choices of time series-to-image transformation in Section 4.3.

**Image Creation.** Given a set of time series line graphs $\mathcal{G}_i = g_1, g_2, \cdots, g_D$ for a time series $\mathcal{S}_i$, we arrange them in a single image $x_i$ using a pre-defined grid layout. We adopt a square grid by default, following [10]. Specifically, we arrange the $D$ time series line graphs in a grid of size $l \times l$ if $l \times (l-1) < D \leq l \times l$, and a grid of size $l \times (l+1)$ if $l \times l < D \leq l \times (l+1)$. For example, the P19, P12, and PAM datasets contain 34, 36, and 17 variables, respectively, and the corresponding default grid layouts are $6 \times 6$, $6 \times 6$, and $4 \times 5$. Any grid spaces not occupied by a line graph remain empty. Figure 6 showcases examples of the resulting images. As for the order of variables, we sort them according to the missing ratios for irregularly sampled time series. We explored the effects of different grid layouts and variable orders in Section 4.3.

## 3.2 Vision Transformers for Time Series Modeling

Given the image $x_i$ transformed from time series $\mathcal{S}_i$, we leverage an image classifier to perceive the image and perform the classification task. The time series patterns in a line graph image involve both local (*i.e.*, the temporal dynamics of a single variable in a line graph) and global (the correlation among variables across different line graphs) contexts. To better capture these patterns, we choose the recently developed vision transformers. Unlike the predominant CNNs, vision transformers are proven to excel at maintaining spatial information and have stronger abilities to capture local and global dependencies [9, 22].

**Preliminary.** Vision Transformer (ViT) [9] is originally adapted from NLP. The input image is split into fixed-sized patches, and each patch is linearly embedded and augmented with position embeddings. The resulting sequence of vectors is then fed into a standard Transformer encoder consisting of a stack of multi-head attention modules (MSA) and MLP to obtain patch representations. An extra classification token is added to the sequence to perform classification or other tasks. ViT models *global* inter-unit interactions between all pairs of patches, which can be computationally expensive for high-resolution images. Swin Transformer, on the other hand, adopts a hierarchical architecture with multi-level feature maps and performs self-attention locally within non-overlapping windows, reducing computational complexity while improving performance. We use Swin Transformer as the default backbone vision model unless otherwise specified, but any other vision model can also be used within this framework.

Swin Transformer captures the *local* and *global* information by constructing a hierarchical representation starting from small-sized patches in earlier layers and gradually merging neighboring patches in deeper layers. Specifically, in the W-MSA block, self-attention is calculated within each non-overlapping window, allowing for the capture of local intra-variable interactions and temporal dynamics of a single line graph for a variable $d$. The shifted window block SW-MSA then enables

connections between different windows, which span across different line graphs, to capture *global* interactions. Fig. 2 illustrates this process. Mathematically, the consecutive Swin Transformer blocks are calculated as:

$$\hat{\mathbf{z}}^l = \text{W-MSA}\left(\text{LN}\left(\mathbf{z}^{l-1}\right)\right) + \mathbf{z}^{l-1},$$
$$\mathbf{z}^l = \text{MLP}\left(\text{LN}\left(\hat{\mathbf{z}}^l\right)\right) + \hat{\mathbf{z}}^l,$$
$$\hat{\mathbf{z}}^{l+1} = \text{SW-MSA}\left(\text{LN}\left(\mathbf{z}^l\right)\right) + \mathbf{z}^l,$$
$$\mathbf{z}^{l+1} = \text{MLP}\left(\text{LN}\left(\hat{\mathbf{z}}^{l+1}\right)\right) + \hat{\mathbf{z}}^{l+1}, \tag{1}$$

where $\hat{\mathbf{z}}^l$ and $\mathbf{z}^l$ denote the output features of the (S)W-MSA module and the MLP module for block $l$, respectively; LN stands for the layer normalization [1]. After multiple stages of blocks, the global interactions among patches from all line graphs can be captured, enabling the modeling of correlations between different variables. We have also explored the use of additional positional embeddings, including local positional embeddings to indicate the position of each patch within its corresponding line graph, and global positional embeddings to represent the index of the associated line graph in the entire image. However, we didn't observe consistent improvement over the already highly competitive performance, which might suggest that the original pre-trained positional embeddings have already been able to capture the information regarding local and global patch positions.

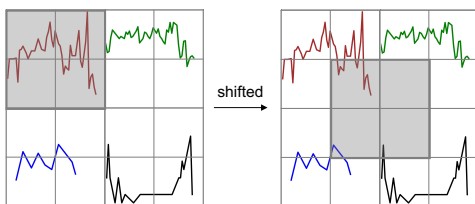

Figure 2: Illustration of the shifted window approach in Swin Transformer. Self-attention is calculated within each window (grey box). When the window is contained within a single line graph, it captures local interactions. After shifting, the window includes patches from different line graphs, allowing for the modeling of global cross-variable interactions.

**Inference.** We use vision transformers to predict the labels of time series in the same way as image classification. The outputs of Swin Transformer blocks at the final stage are used as the patch representations, upon which a flattened layer with a linear head is applied to obtain the prediction $\hat{y}_i$.

## 4 Experiments

### 4.1 Experimental Setup

Table 1: Statistics of the irregularly sampled time series datasets used in our experiments.

| Datasets | #Samples | #Variables | #Avg. obs. | #Classes | Demographic info | Imbalanced | Missing ratio |
|---|---|---|---|---|---|---|---|
| P19 | 38,803 | 34 | 401 | 2 | True | True | 94.9% |
| P12 | 11,988 | 36 | 233 | 2 | True | True | 88.4% |
| PAM | 5,333 | 17 | 4,048 | 8 | False | False | 60.0% |

**Datasets and metrics.** We conducted experiments using three widely used healthcare and human activity datasets, as outlined in Table 1. The P19 dataset [29] contains information from 38,803 patients, with 34 sensor variables and a binary label indicating sepsis. The P12 dataset [12] comprises data from 11,988 patients, including 36 sensor variables and a binary label indicating survival during hospitalization. Lastly, the PAM dataset [28] includes 5,333 samples from 8 distinct human activities, with 17 sensor variables provided for each sample. We used the processed data provided by Raindrop [49]. To ensure consistency, we employed identical data splits across all comparison baselines, and evaluated the performance using standard metrics such as Area Under a ROC Curve (AUROC) and Area Under Precision-Recall Curve (AUPRC) for the imbalanced P12 and P19 datasets. For the more balanced PAM dataset, we reported Accuracy, Precision, Recall, and F1 score.

**Implementation.** We utilized the Matplotlib package to plot line graphs and save them as standard RGB images. For the P19, P12, and PAM datasets, we implemented grid layouts of $6 \times 6$, $6 \times 6$, and $4 \times 5$, respectively. For a fair comparison, we assigned a fixed size of $64 \times 64$ to each grid cell (line graph), resulting in image sizes of $384 \times 384$, $384 \times 384$, and $256 \times 320$, respectively. It is important to note that image sizes can also be directly set to any size, irrespective of grid cell dimensions. We plot the images according to the value scales on training sets. We employed the checkpoint of Swin

Table 2: Comparison with the baseline methods on irregularly sampled time series classification task. **Bold** indicates the best performer, while underline represents the second best.

| Methods | P19 | | P12 | | PAM | | | |
| --- | --- | --- | --- | --- | --- | --- | --- | --- |
| | AUROC | AUPRC | AUROC | AUPRC | Accuracy | Precision | Recall | F1 score |
| Transformer | $80.7 \pm 3.8$ | $42.7 \pm 7.7$ | $83.3 \pm 0.7$ | $47.9 \pm 3.6$ | $83.5 \pm 1.5$ | $84.8 \pm 1.5$ | $86.0 \pm 1.2$ | $85.0 \pm 1.3$ |
| Trans-mean | $83.7 \pm 1.8$ | $45.8 \pm 3.2$ | $82.6 \pm 2.0$ | $46.3 \pm 4.0$ | $83.7 \pm 2.3$ | $84.9 \pm 2.6$ | $86.4 \pm 2.1$ | $85.1 \pm 2.4$ |
| GRU-D | $83.9 \pm 1.7$ | $46.9 \pm 2.1$ | $81.9 \pm 2.1$ | $46.1 \pm 4.7$ | $83.3 \pm 1.6$ | $84.6 \pm 1.2$ | $85.2 \pm 1.6$ | $84.8 \pm 1.2$ |
| SeFT | $81.2 \pm 2.3$ | $41.9 \pm 3.1$ | $73.9 \pm 2.5$ | $31.1 \pm 4.1$ | $67.1 \pm 2.2$ | $70.0 \pm 2.4$ | $68.2 \pm 1.5$ | $68.5 \pm 1.8$ |
| mTAND | $84.4 \pm 1.3$ | $50.6 \pm 2.0$ | $84.2 \pm 0.8$ | $48.2$$\pm 3.4$ | $74.6 \pm 4.3$ | $74.3 \pm 4.0$ | $79.5 \pm 2.8$ | $76.8 \pm 3.4$ |
| IP-Net | $84.6 \pm 1.3$ | $38.1 \pm 3.7$ | $82.6 \pm 1.4$ | $47.6 \pm 3.1$ | $74.3 \pm 3.8$ | $75.6 \pm 2.1$ | $77.9 \pm 2.2$ | $76.6 \pm 2.8$ |
| DGM$^2$-O | $86.7 \pm 3.4$ | $44.7 \pm 11.7$ | $84.4$$\pm 1.6$ | $47.3 \pm 3.6$ | $82.4 \pm 2.3$ | $85.2 \pm 1.2$ | $83.9 \pm 2.3$ | $84.3 \pm 1.8$ |
| MTGNN | $81.9 \pm 6.2$ | $39.9 \pm 8.9$ | $74.4 \pm 6.7$ | $35.5 \pm 6.0$ | $83.4 \pm 1.9$ | $85.2 \pm 1.7$ | $86.1 \pm 1.9$ | $85.9 \pm 2.4$ |
| Raindrop | $87.0$$\pm 2.3$ | $51.8$$\pm 5.5$ | $82.8 \pm 1.7$ | $44.0 \pm 3.0$ | $88.5$$\pm 1.5$ | $89.9$$\pm 1.5$ | $89.9$$\pm 0.6$ | $89.8$$\pm 1.0$ |
| **ViTST** | **$89.2$**$\pm 2.0$ | **$53.1$**$\pm 3.4$ | **$85.1$**$\pm 0.8$ | **$51.1$**$\pm 4.1$ | **$95.8$**$\pm 1.3$ | **$96.2$**$\pm 1.3$ | **$96.1$**$\pm 1.1$ | **$96.5$**$\pm 1.2$ |

Transformer pre-trained on the ImageNet-21K dataset[2]. The default patch size and window size are 4 and 7, respectively.

**Training.** Given the highly imbalanced nature of the P12 and P19 datasets, we employed upsampling of the minority class to match the size of the majority class. We finetuned the Swin Transformer for 2 and 4 epochs on the upsampled P19 and P12 datasets, respectively, and for 20 epochs on the PAM dataset. The batch sizes used for training were 48 for P19 and P12, and 72 for PAM, while the learning rate was set to 2e-5. The models were trained using A6000 GPUs with 48G memory.

**Incorporating static features.** In real-world applications, especially in the healthcare domain, irregular time series data often accompanies additional information such as categorical or textual features. In the P12 and P19 datasets, each patient's demographic information, including weight, height, and ICU type, is provided. This static information remains constant over time and can be expressed using natural language. To incorporate this information into our framework, we employed a template to convert it into natural language sentences and subsequently encoded the resulting text using a RoBERTa-base [21] text encoder. The resulting text embeddings were concatenated with the image embeddings obtained from the vision transformer to perform classification. The static features were also applied to all compared baselines.

## 4.2 Main Results

**Comparison to state-of-the-art.** We compare our approach with several state-of-the-art methods that are specifically designed for irregularly sampled time series, including Transformer [38], Trans-mean (Transformer with an imputation method that replaces the missing value with the average observed value of the variable), GRU-D [3], SeFT [16], mTAND [34], IP-Net [33], and Raindrop [49]. In addition, we also compared our method with two approaches initially designed for forecasting tasks, namely DGM$^2$-O [42] and MTGNN [43]. The implementation and hyperparameter settings of these baselines were kept consistent with those used in Raindrop [49]. Specifically, a batch size of 128 was employed, and all compared models were trained for 20 epochs. To ensure fairness in our evaluation, we averaged the performance of each method over 5 data splits that were kept consistent across all compared approaches.

As shown in Table 2, our proposed approach demonstrates strong performance against the specialized state-of-the-art algorithms on all three datasets. Specifically, on the P19 and P12 datasets, ViTST achieves improvements of 2.2% and 0.7% in absolute AUROC points, and 1.3% and 2.9% in absolute AUPRC points over the state-of-the-art results, respectively. For the PAM dataset, the improvement is even more significant, with an increase of 7.3% in Accuracy, 6.3% in Precision, 6.2% in Recall, and 6.7% in absolute F1 score points.

**Leaving-sensors-out settings.** We conducted additional evaluations to assess the performance of our model under more challenging scenarios, where the observations of a subset of sensors (variables) were masked during testing. This setting simulates real-world scenarios when some sensors fail or become unreachable. Following the approach adopted in [49], we experimented with two setups using the PAM dataset: (1) *leave-**fixed**-sensors-out*, which drops a fixed set of sensors across all

---

[2]https://huggingface.co/microsoft/swin-base-patch4-window7-224-in22k

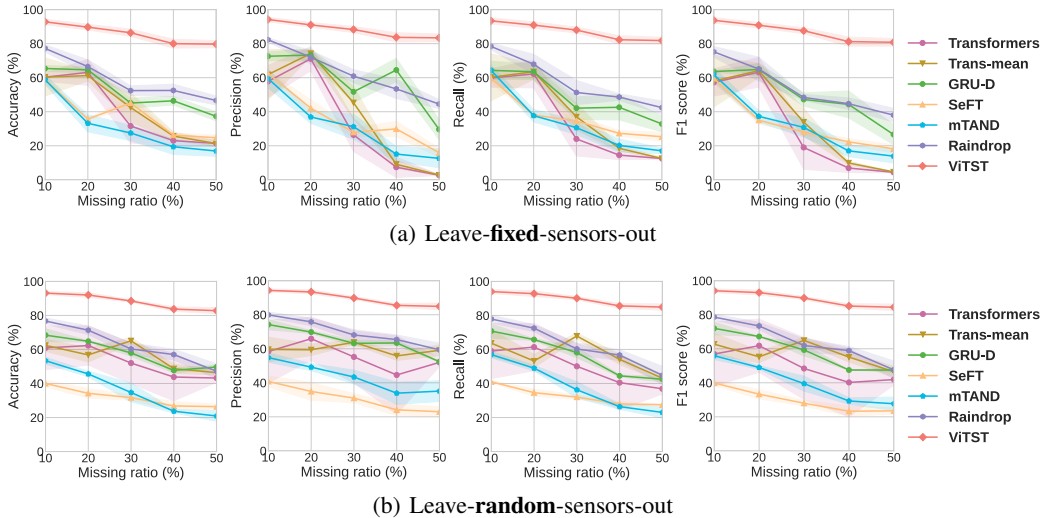

(a) Leave-**fixed**-sensors-out

(b) Leave-**random**-sensors-out

Figure 3: Performance in leave-**fixed**-sensors-out and leave-**random**-sensors-out settings on PAM dataset. The x-axis is the "missing ratio" which denotes the ratio of masked variables.

samples and compared methods, and (2) *leave-**random**-sensors-out*, which drops sensors randomly. It is important to note that only the observations in the validation and test sets were dropped, while the training set was kept unchanged. To ensure a fair comparison, we dropped the same set of sensors in the *leave-**fixed**-sensors-out* setting as in [49].

The results are presented in Fig. 3, from which we observe that our approach consistently achieves the best performance and outperforms all the specialized baselines by a large margin. With the missing ratio increasing from 10% to 50%, our approach maintains strong performance, staying above 80%. In contrast, the comparison baseline shows a marked drop. The advantage of ViTST over the comparison baselines becomes increasingly significant. Even when half of the variables were dropped, our approach was still able to achieve acceptable performance over 80, surpassing the best-performing baseline Raindrop by 33.1% in Accuracy, 40.9% in Precision, 39.4% in Recall, and 42.8% in F1 score in the *leave-**fixed**-sensors-out* setting (all in absolute points). We also notice that the variances in our results are notably lower compared to the baselines. These results suggest that our approach is highly robust to varying degrees of missing observations in time series.

## 4.3 Additional Analysis

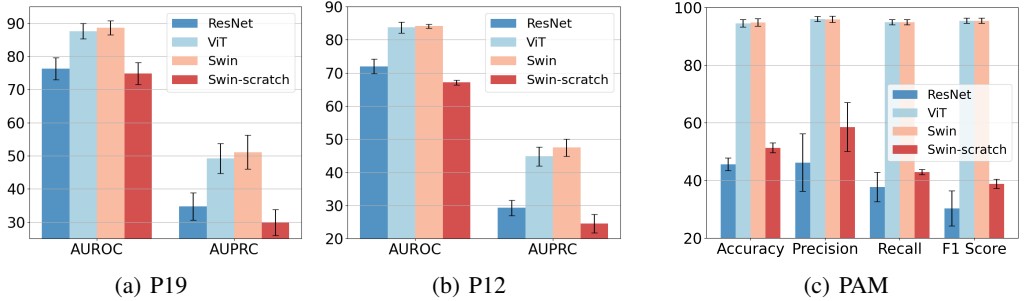

(a) P19

(b) P12

(c) PAM

Figure 4: Performance of different backbone vision models on P19, P12, and PAM datasets. We do not use static features for our approach here to exclude their influence.

**Where does the performance come from?** Our approach transforms time series into line graph images, allowing the use of vision transformers for time series modeling. We hypothesize that vision transformers can leverage their general-purpose image recognition ability acquired from large-scale pre-training on natural images (such as ImageNet [30]) to capture informative patterns in the line graph images. To validate that, we compared the performance of a pre-trained Swin

Table 3: Ablation studies on different strategies of time series-to-image transformation.

| Methods | P19 | | P12 | | PAM | | | |
|---|---|---|---|---|---|---|---|---|
| | AUROC | AUPRC | AUROC | AUPRC | Accuracy | Precision | Recall | F1 score |
| Default | $89.2 \pm 2.0$ | $53.1 \pm 3.4$ | $85.1 \pm 0.8$ | $51.1 \pm 4.1$ | $95.8 \pm 1.3$ | $96.2 \pm 1.1$ | $96.2 \pm 1.3$ | $96.5 \pm 1.2$ |
| w/o interpolation | $89.6 \pm 2.1$ | $52.9 \pm 3.4$ | $85.7 \pm 1.0$ | $51.9 \pm 3.4$ | $95.6 \pm 1.1$ | $96.6 \pm 0.9$ | $95.9 \pm 1.0$ | $96.2 \pm 1.0$ |
| w/o markers | $89.0 \pm 2.1$ | $51.7 \pm 2.5$ | $85.3 \pm 0.9$ | $50.3 \pm 3.2$ | $95.8 \pm 1.1$ | $96.9 \pm 0.7$ | $96.0 \pm 1.0$ | $96.4 \pm 0.9$ |
| w/o colors | $88.8 \pm 1.8$ | $51.4 \pm 4.1$ | $84.4 \pm 0.7$ | $47.0 \pm 2.9$ | $95.0 \pm 1.0$ | $96.2 \pm 0.7$ | $95.3 \pm 1.0$ | $95.7 \pm 0.9$ |
| w/o order | $89.3 \pm 2.3$ | $52.7 \pm 4.5$ | $84.0 \pm 1.8$ | $47.8 \pm 4.6$ | - | - | - | - |

Transformer with a Swin Transformer trained from scratch, as shown in Fig. 4. The significant drop in performance without pre-training provides evidence that Swin transformer could transfer the knowledge obtained from pre-training on natural images to our synthetic time series line graph images, achieving impressive performance. Nevertheless, the underlying mechanism needs further exploring and probing in future studies.

**How do different vision models perform?** We benchmarked several backbone vision models within our framework. Specifically, we tried another popular pre-trained vision transformer ViT[3] and a pre-trained CNN-based model, ResNet[4]. The results are presented in Fig. 4. The pre-trained transformer-based ViT and Swin Transformer demonstrate comparable performance, both outperforming the previous state-of-the-art method, Raindrop. In contrast, the pre-trained CNN-based ResNet lagged considerably behind the vision transformer models. This performance gap is consistent with observations in image classification tasks on datasets like ImageNet, where vision transformers have been shown to excel in preserving spatial information compared to conventional CNN models. This advantage enables vision transformers to effectively capture the positions of patches within each line graph sub-image and the entire image and facilitates the modeling of complex dynamics and relationships between variables.

**How to create time series line graph images?** Using line graphs to visualize time series offers us an intuitive way to interpret the data and adjust the visualization strategy for enhanced clarity and potentially augment the performance. To offer insights on effective time series-to-image transformation, we analyze the effects of several key elements in practice: (1) the default linear *interpolation* to connect partially observed data points on the line graphs; (2) *markers* to indicate observed data points; (3) variable-specific *colors* to differentiate between line graphs representing different variables; (4) the *order* determined by missing ratios when organizing multiple line graphs in a single image.

The results are presented in Table 3. Given the balanced missing ratios in the PAM dataset, we excluded results without the sorting order. Interestingly, plotting only observed data points without linear interpolation led to better results on P19 and P12 datasets. This could be attributed to the potential inaccuracies introduced by interpolation, blurring distinctions between observed and interpolated points. Additionally, omitting markers complicates the model's task of discerning observed data points from interpolated ones, degrading its performance. The absence of distinctive colors led to the most significant performance drop, underlining the necessity of using varied hues for individual line graphs to help the model to distinguish them. While a specific sorting order may not ensure optimal outcomes across all datasets, it does provide relatively stable results over multiple datasets and evaluation metrics. For the PAM dataset, these nuances seem to have minimal impact, indicating the robustness of our approach against these variations in some scenarios.

**Effects of grid layouts and image sizes.** We explored the influence of grid layouts and image dimensions on our approach's efficacy. For a fair comparison across grid layouts, we fixed the size of each grid cell as $64 \times 64$ and altered the grid layouts. As shown in Figure 5. we observed our approach's robustness to variations in grid layouts, with square layout consistently yielding good results across different datasets and metrics, which was particularly evident for the P12 dataset. Regarding image size, when we maintained the grid layout but reduced the overall image dimensions, a noticeable performance decline was observed on the P12 and PAM datasets, which complies with our intuition.

**Robustness against varied plotting parameters.** To gauge the robustness of our approach against different plotting parameters, we assessed aspects including line style/width and marker style/size,

---

[3]https://huggingface.co/google/vit-base-patch16-224-21k
[4]https://huggingface.co/microsoft/resnet-50

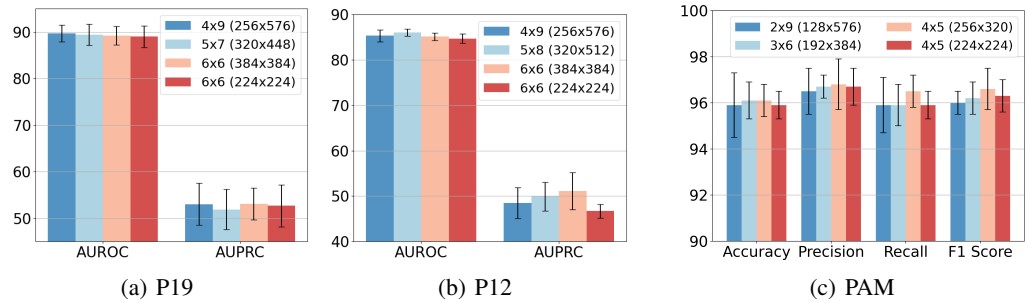

(a) P19  (b) P12  (c) PAM

Figure 5: Ablation study of the influence of grid layouts and image sizes. For instance, 4x9 (256x576) denotes a grid layout of 4×9 with an image size of 256×576 pixels.

primarily using the P19 dataset. As shown in Table 4, our approach demonstrates robustness against changes in these parameters, maintaining strong performance across different plotting configurations.

**What does ViTST capture?** To gain insights into the patterns captured by ViTST in the time series line graph images, we analyzed the averaged attention map of a ViTST model with ViT as the backbone, as depicted in Fig. 6. The attention map reveals that the model consistently attends to the informative part, i.e., line graph contours within the image. Furthermore, we observed that the model appropriately focuses on observed data points and areas where the slopes of the lines change. Conversely, flat line graphs that lack dynamic patterns seem to receive less attention. This demonstrates that ViTST might be able to discern between informative and un-informative features in the line graph images, enabling it to extract meaningful patterns.

Table 4: Robustness regarding the style and size of lines and markers. In the brackets, the first element denotes style, and the second represents size.

| Line | Marker | AUROC | AUPRC |
|------|--------|-------|-------|
| (solid,1) | $(*, 2)$ | $89.2 \pm 2.0$ | $53.1 \pm 3.4$ |
| (dashed,1) | $(*, 2)$ | $89.2 \pm 2.1$ | $53.7 \pm 4.1$ |
| (dotted,1) | $(*, 2)$ | $89.2 \pm 2.1$ | $52.8 \pm 4.0$ |
| (solid,0.5) | $(*, 2)$ | $88.6 \pm 1.7$ | $53.0 \pm 3.6$ |
| (solid,1) | $(*, 2)$ | $89.2 \pm 2.0$ | $53.1 \pm 3.4$ |
| (solid,2) | $(*, 2)$ | $88.5 \pm 2.3$ | $53.6 \pm 3.1$ |
| (solid,1) | $(*, 2)$ | $89.2 \pm 2.0$ | $53.1 \pm 3.4$ |
| (solid,1) | $(\wedge, 2)$ | $89.3 \pm 1.9$ | $52.6 \pm 4.0$ |
| (solid,1) | $(\circ, 2)$ | $89.1 \pm 1.9$ | $51.3 \pm 4.2$ |
| (solid,1) | $(*, 1)$ | $88.2 \pm 1.4$ | $52.1 \pm 4.5$ |
| (solid,1) | $(*, 2)$ | $89.2 \pm 2.0$ | $53.1 \pm 3.4$ |
| (solid,1) | $(*, 3)$ | $88.9 \pm 1.9$ | $52.8 \pm 3.2$ |

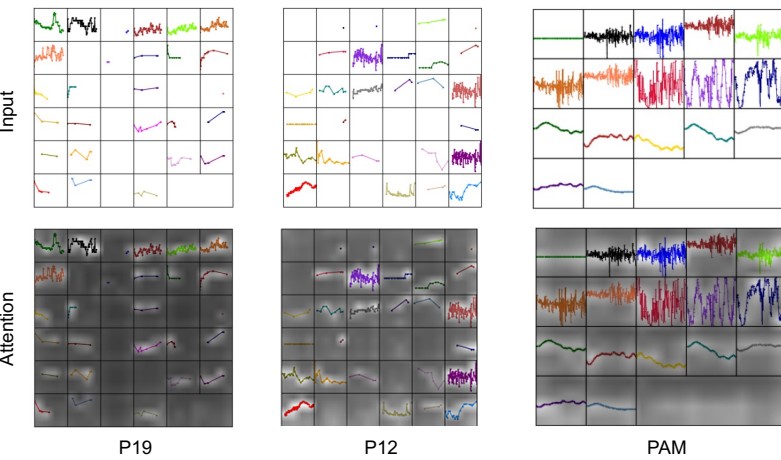

P19  P12  PAM

Figure 6: Illustration of the averaged attention map of ViTST.

## 4.4 Regular Time Series Classification

An advantage of our approach is its ability to model time series of diverse shapes and scales, whether they are regular or irregular. To evaluate the performance of our approach on regular time series data,

Table 5: Performance comparison on regular multivariate time series datasets. **Bold** indicates the best performer, while underline represents the second best.

| Datasets | EC | UW | SCP1 | SCP2 | JV | SAD | HB | FD | PS | EW | Average |
|---|---|---|---|---|---|---|---|---|---|---|---|
| *Dataset statistics* | | | | | | | | | | | |
| #Variables | 3 | 3 | 6 | 7 | 12 | 13 | 61 | 144 | **963** | 6 | - |
| Length | 1,751 | 315 | 896 | 1,152 | 29 | 93 | 405 | 62 | 144 | **17984** | - |
| *Model performances* | | | | | | | | | | | |
| $DTW_D$ | 0.323 | 0.903 | 0.775 | 0.539 | 0.949 | 0.963 | 0.717 | 0.529 | 0.711 | 0.618 | 0.717 |
| LSTM | 0.323 | 0.412 | 0.689 | 0.466 | 0.797 | 0.319 | 0.722 | 0.577 | 0.399 | - | 0.523 |
| XGBoost | 0.437 | 0.759 | 0.846 | 0.489 | 0.865 | 0.696 | 0.732 | 0.633 | **0.983** | - | 0.727 |
| Rocket | 0.452 | **0.944** | 0.908 | 0.533 | 0.962 | 0.712 | 0.756 | 0.647 | 0.751 | - | 0.741 |
| TST | 0.326 | 0.913 | **0.922** | **0.604** | **0.997** | **0.998** | **0.776** | **0.689** | 0.896 | 0.748 | **0.791** |
| ViTST | **0.456** | 0.862 | 0.898 | 0.561 | 0.946 | 0.985 | 0.766 | 0.632 | 0.913 | **0.878** | 0.780 |

we conducted experiments on ten representative multivariate time series datasets from the UEA Time Series Classification Archive [2]. These datasets exhibit diverse characteristics, as summarized in Table 5. It is worth noting that the PS dataset in our evaluation contains an exceptionally high number of variables (963), while the EW dataset has extremely long time series (17984). We specifically selected these two datasets to assess the effectiveness of our approach in handling large numbers of variables and long time series. We follow [47] to use these baselines for comparison: $DTW_D$ which stands for dimension-Dependent DTW combined with dilation-CNN [11], LSTM [13], XGBoost [5], Rocket [7], and a transformer-based TST [47] which operates on fully observed numerical time series.

The performance of our approach on regular time series datasets is consistently strong, as demonstrated in Table 5. With an average accuracy that is second-best and closely aligned with the top-performing baseline method TST, our approach showcases its competitive capabilities. Notably, it excels on challenging datasets PS and EW with massive variables and observation length. These results were achieved using the same image resolution ($384 \times 384$) as the other datasets, indicating the effectiveness and efficiency of our approach. The ability of our approach to handle both irregular and regular time series data further emphasizes its versatility and broad applicability.

# 5   Conclusion

In this paper, we introduce a novel perspective on modeling irregularly sampled time series. By transforming time series data into line graph images, we could effectively leverage the strengths of pre-trained vision transformers. This approach is straightforward yet effective and versatile, enabling the modeling of time series of diverse characteristics, regardless of irregularity, different structures, and scales. Through extensive experiments, we demonstrate that our approach surpasses state-of-the-art methods designed for irregular time series and maintains strong robustness to varying degrees of missing observations. Additionally, our approach achieves promising results on regular time series data. We envision its potential as a general-purpose framework for various time series tasks. Our results underscore the potential of adapting rapidly advancing computer vision techniques to time series modeling.

# 6   Limitations and Future Work

In this work, we utilized a straightforward method to image multivariate time series by converting them into line graph images using matplotlib and then saving them as RGB images. While our results are promising and exhibit robustness against variations in the time series-to-image transformation process, there may be alternative ways to visualize the data. This includes potentially more controllable and accurate plotting methods or different image representations beyond line graphs. Our findings also highlight the efficacy of pre-trained vision transformers for time series classification, suggesting that these models might leverage knowledge acquired from pre-training on natural images. Yet, the underlying reasons for their remarkable success still need deeper exploration and investigation. This research serves as a promising starting point in this domain, suggesting various potential directions. We leave these further explorations and investigations for future work.

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

# A  More Details on Time Series Line Graph Image Creation

**Implementation**  The time-series-to-image transformation can be implemented using the Matplotlib package[5] with the following few lines of code.

```python
def TS2Image(t, v, D, colors, image_height, image_width, grid_height, grid_width):
    import matplotlib.pyplot as plt
    plt.figure(figsize=(image_height/100, image_width/100), dpi=100)
    for d in range(D): # enumerate the multiple variables
        plt.subplot(grid_height, grid_width, d+1) # position in the grid
        # plot line graph of variable d
        plt.plot(t[d], v[d], color=colors[d], linestyle="-", marker="*")
```

**Axis Limits of Line Graphs**  The axis limits determine the plot area of the line graphs and the range of displayed timestamps and values. By default, we set the limits of the x-axis and y-axis as the ranges of all the observed timestamps and values. However, we found that some extreme observed values for some variables can largely expand the range of the y-axis, causing most plotted points of observations to cluster in a small area and resulting in flat line graphs. Common normalization and standardization methods will not solve this issue, as the relative magnitudes remain unchanged in the created images. We thus tried the following strategies to remove extreme values and narrow the range of the y-axis in our preliminary experiments:

- Interquartile Range (IQR): IQR is one of the most extensively used methods for outlier detection and removal. The interquartile range is calculated based on the first and third quartiles of all the observed values of each variable in the dataset and then used to calculate the upper and lower limits.

- Standard Deviation (SD): The upper and lower boundaries are calculated by taking 3 standard deviations from the mean of observed values for each variable across the dataset. This method usually assumes the data is normally distributed.

- Modified Z-score (MZ): A z-score measures how many standard deviations away a value is from the mean and is similar to the standard deviation method to detect outliers. However, z-scores can be influenced by extreme values, which modified z-scores can better handle. We set the upper and lower limits as the values whose modified z-scores are 3.5 and -3.5.

We show examples of the created images with these strategies in Figure 7.

Table 6: Preliminary experiments on different strategies to decide the line graph limit. The default strategy is to directly set the axis limit as the range of all observed values on the dataset. "IQR", "SD", and "MZS' denote three strategies to remove extreme value, *i.e.,* Interqurtile Range, Standard Deviation, and Modified Z-score. The reported numbers are averaged on 5 data splits.

| Strategies | P19 | | P12 | | PAM | | | |
|---|---|---|---|---|---|---|---|---|
| | AUROC | AUPRC | AUROC | AUPRC | Accuracy | Precision | Recall | F1 score |
| Default | $89.4 \pm 1.9$ | $52.8 \pm 3.8$ | $85.6 \pm 1.1$ | $49.8 \pm 2.5$ | $96.1 \pm 0.7$ | $96.8 \pm 1.1$ | $96.5 \pm 0.7$ | $96.6 \pm 0.9$ |
| IQR | $88.2 \pm 0.8$ | $49.6 \pm 1.7$ | $84.5 \pm 1.1$ | $48.9 \pm 2.6$ | $95.9 \pm 0.7$ | $96.8 \pm 0.7$ | $96.1 \pm 0.7$ | $96.4 \pm 0.7$ |
| SD | $87.4 \pm 1.6$ | $51.2 \pm 3.6$ | $84.6 \pm 1.7$ | $47.1 \pm 2.9$ | $96.6 \pm 0.9$ | $97.1 \pm 0.8$ | $97.0 \pm 0.6$ | $97.0 \pm 0.7$ |
| MZS | $87.3 \pm 1.0$ | $50.8 \pm 3.7$ | $84.3 \pm 1.4$ | $47.1 \pm 2.1$ | $96.0 \pm 1.1$ | $96.8 \pm 0.99$ | $96.4 \pm 0.9$ | $96.6 \pm 0.9$ |

The performance comparison of models trained on images created with different strategies is shown in Table 6. We observe that the methods that remove extreme values hurt the performance, except for SD on the PAM dataset. Although these methods narrow the value range and highlight the dynamic patterns of line graphs, they discard the extreme values that might be informative themselves. Therefore, we stick with the default way that sets the axis limits as the range of all observed values.

**Computation cost**  We list the inference time (in seconds) of different methods on the test sets of three datasets in Table 7. All the inferences are made in a single Nvidia A6000 GPU. It is observed that our vision-based method consumes more inference time than the non-vision baselines. However, we believe this cost remains within an acceptable range in the context of today's ML practice and medical applications, considering the inference on each sample only costs around 0.01 seconds.

---

[5]https://matplotlib.org/

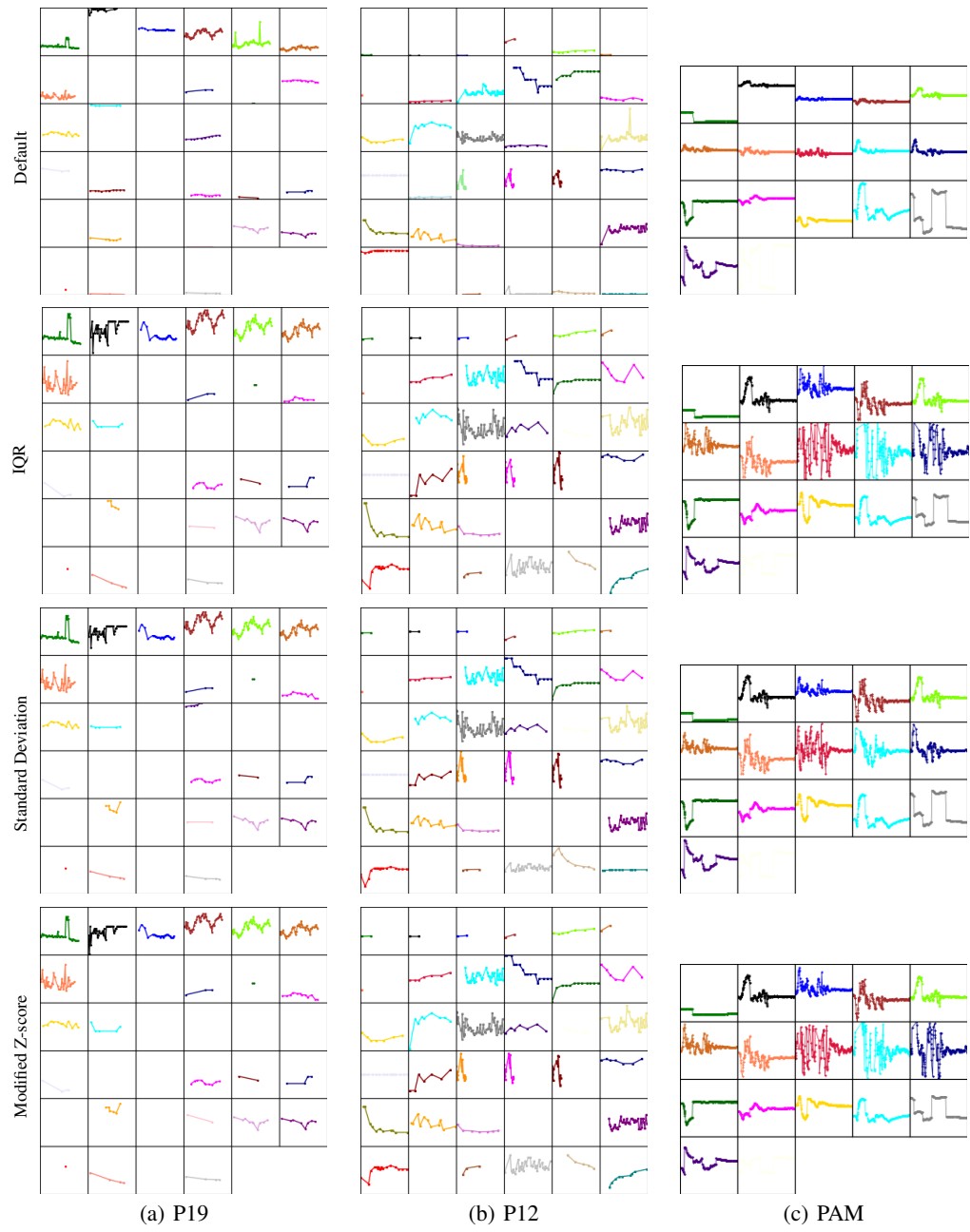

(a) P19     (b) P12     (c) PAM

Figure 7: The images created with different strategies for three samples from P19, P12, and PAM dataset, respectively (sample "p000019" for P19, "132548" for P12, and "0" for PAM).

Table 7: The inference time (in seconds) of different methods on the test sets.

| Datasets | Trasformers | mTAND | SeFT | Raindrop | MTGNN | DGM$^2$-O | GRU-D | ViTST |
|---|---|---|---|---|---|---|---|---|
| P19 | 0.21 | 0.52 | 2.72 | 3.05 | 3.62 | 2.47 | 31.04 | 44.51 |
| P12 | 0.12 | 0.44 | 0.97 | 1.27 | 1.46 | 2.80 | 10.13 | 12.14 |
| PAM | 0.06 | 0.23 | 0.89 | 0.67 | 1.16 | 2.98 | 4.55 | 5.30 |

## B  More Experimental Details

### B.1  Datasets

We used the datasets processed by [49], whose details are given below.

**P19: PhysioNet Sepsis Early Prediction Challenge 2019.** [6] The P19 dataset [29] consists of clinical data for 38,803 patients, and aims to predict whether sepsis will occur within the next 6 hours. The dataset includes 34 irregularly sampled sensors with 8 vital signs and 26 laboratory values for each patient, as well as 6 demographic features. To process the static features, we use templates outlined in Table 8, and utilize a pre-trained Roberta-base model to extract textual features. These textual features are then combined with visual features obtained from the vision transformer to perform binary classification. The dataset is highly imbalanced with only 4% of samples being positive, and has a missing ratio of 94.9%.

**P12: PhysioNet Mortality Prediction Challenge 2012.** [7] P12 dataset [12] includes clinical data from 11,988 ICU patients, with 36 irregularly sampled sensor observations and 6 static demographic features provided for each patient. The goal is to predict patient mortality, which is a binary classification task. The dataset is highly imbalanced, with around 86% of samples being negative. The missing ratio of the dataset is 88.4%.

**PAM: PAMAP2 Physical Activity Monitoring.** [8] The PAM dataset originally contains data of 18 physical activities with 9 subjects wearing 3 inertial measurement units. However, to make it suitable for irregular time series classification, [49] excluded the ninth subject due to its short length of sensor readouts, and 10 out of the 18 activities that had less than 500 samples were also excluded. As a result, the task is an 8-way classification with 5,333 samples, each with 600 continuous observations. To simulate the irregular time series setting, 60% of the observations are randomly removed. No static features are provided, and the 8 categories are approximately balanced. The missing ratio is 60.0%.

Table 8: Templates for transforming static features to natural language sentences.

| Dataset | Static features | Template | Example |
|---|---|---|---|
| P19 | *Age*, *Gender*, *Unit1 (medical ICU)*, *Unit2 (surgery ICU)*, *HospAdmTime*; *ICULOS (ICU length-of-stay)* | A patient is {*Age*} years old, {*Gender*}, went to {*Unit1 & Unit2*} {*HospAdmTime*} hours after hospital admit, had stayed there for {*ICULOS*} hours. | A patient is 65 years old, female, went to the medical ICU 10 hours after hospital admit, had stayed there for 20 hours. |
| P12 | *RecordID*, *Age*, *Gender*, *Height* (cm), *ICUType*, *Weight* (kg) | A patient is {*Age*} years old, {*Gender*}, {*Height*} cm, {*Weight*} kg, stayed in {*ICUType*}. | A patient is 48 years old, male, 171 cm, 78 kg, stayed in surgical ICU. |

Table 9: Ablation studies on different methods to encode static features.

| Methods | P19 | | P12 | |
|---|---|---|---|---|
| | AUROC | AUPRC | AUROC | AUPRC |
| Raindrop | $87.0 \pm 2.3$ | $51.8 \pm 5.5$ | $82.8 \pm 1.7$ | $44.0 \pm 3.0$ |
| Swin | $89.4 \pm 1.8$ | $50.2 \pm 3.0$ | $84.3 \pm 0.6$ | $49.3 \pm 3.7$ |
| Swin-MLP | $88.6 \pm 1.3$ | $51.4 \pm 3.7$ | $84.6 \pm 0.9$ | $48.7 \pm 3.2$ |
| Swin-Roberta | $89.4 \pm 1.9$ | $52.8 \pm 3.8$ | $85.6 \pm 1.1$ | $49.8 \pm 2.5$ |

### B.2  Experiments on Static Features

Time series data is often associated with information from other modalities, such as the textual clinical notes in electronic health records (EHRs) in the healthcare domain. Our approach is naturally suitable for incorporating such information since we convert time series data to images, and thus various vision-language and multi-modal techniques can be utilized to incorporate the visual (time series) information and information from other modalities. For example, the CLIP [27] learns a shared

---

[6]https://physionet.org/content/challenge-2019/1.0.0/
[7]https://physionet.org/content/challenge-2012/1.0.0/
[8]https://archive.ics.uci.edu/ml/datasets/pamap2+physical+activity+monitoring

hidden feature space where the paired image and text stay close. Under our framework, such a shared space can also be learned for the paired visual time series images and textual clinical notes, which is our future direction. It also paves the way for the application of multi-modal models such as GPT-4 [26] to handle the visualized time series data and the clinical notes simultaneously. In our current experiments, we used a text encoder, Roberta-base, to encode textual demographic information in the P19 and P12 datasets. We also experimented with normalizing the original categorical features and encoding them using an MLP as in previous work, and compare with the strong baseline, Raindrop. The results are shown in Table 9. We observe that even without using static features, our method has already outperformed Raindrop. In addition, utilizing Roberta to encode and incorporate the textual feature is more effective than applying MLP over categorical features.

Table 10: Preliminary experiments on two settings to fuse the line graph images for different variables. The default setting is to first arrange all line graph images into a single image and then learn the representation for classification. **ViT-subimage** stands for that we first learn the representation for each line graph subimage separately and then concatenate their representation for classification.

| Strategies | P19 | | P12 | | PAM | | | |
|---|---|---|---|---|---|---|---|---|
| | AUROC | AUPRC | AUROC | AUPRC | Accuracy | Precision | Recall | F1 score |
| ViT | $87.9 \pm 2.5$ | $51.6 \pm 3.7$ | $84.8 \pm 1.3$ | $48.1 \pm 3.8$ | $93.4 \pm 0.7$ | $94.7 \pm 0.9$ | $94.1 \pm 0.7$ | $94.3 \pm 0.7$ |
| ViT-subimage | $85.1 \pm 1.5$ | $47.9 \pm 3.4$ | $77.6 \pm 3.2$ | $35.5 \pm 6.2$ | $90.4 \pm 1.3$ | $92.9 \pm 1.0$ | $91.1 \pm 0.9$ | $91.9 \pm 1.0$ |

## B.3 Experiments on Image Fusion

In our preliminary experiments, we examined two distinct approaches to fusing the line graph sub-images in each multivariate time series data. First, we processed each sub-image independently to learn the patch/image representations and then concatenated their respective patch representations to input into the final prediction layer. In contrast, our default method aggregates all sub-images into a single image to learn the patch embeddings. The key distinction between these strategies is whether a patch can attend to those from other sub-images and how position embedding factors in during the representation learning phase. Aside from this, other parameters were consistent across both methods. We tested on ViT and the performance comparisons with these two settings are shown in Table 10. It is evidenced that attending to patches from other sub-images offers advantages. This likely allows for capturing cross-variable correlations at a granular level within the self-attention layers, as opposed to only at the final linear prediction layer.

Table 11: Statistics and hyperparameter settings of evaluated regular multivariate time series datasets.

| Datasets | Variables | Classes | Length | Train size | Grid layout | Image size | Learning rate | Epochs |
|---|---|---|---|---|---|---|---|---|
| EC | 3 | 4 | 1,751 | 261 | $2 \times 2$ | $256 \times 256$ | 1e-4 | 20 |
| UW | 3 | 8 | 315 | 120 | $2 \times 2$ | $256 \times 256$ | 1e-4 | 100 |
| SCP1 | 6 | 2 | 896 | 268 | $2 \times 3$ | $256 \times 384$ | 1e-4 | 100 |
| SCP2 | 7 | 2 | 1,152 | 200 | $3 \times 3$ | $384 \times 384$ | 5e-5 | 100 |
| JV | 12 | 9 | 29 | 270 | $4 \times 4$ | $384 \times 384$ | 1e-4 | 100 |
| SAD | 13 | 10 | 93 | 6599 | $4 \times 4$ | $384 \times 384$ | 1e-5 | 20 |
| HB | 61 | 2 | 405 | 204 | $4 \times 4$ | $384 \times 384$ | 1e-4 | 100 |
| FD | 144 | 2 | 62 | 5890 | $12 \times 12$ | $384 \times 384$ | 5e-4 | 100 |
| PS | **963** | 7 | 144 | 267 | $32 \times 32$ | $384 \times 384$ | 5e-4 | 100 |
| EW | 6 | 5 | **17984** | 128 | $2 \times 3$ | $256 \times 384$ | 2e-5 | 100 |

## B.4 Experiment on Regular Time Series

We selected ten representative multivariate time series datasets from the UEA Time Series Classification Archive [2] with diverse characteristics, including the number of classes, variables, and time series length. The datasets we chose are EthanolConcentration (EC), Handwriting (HW), UWaveGestureLibrary (UW), SelfRegulationSCP1 (SCP1), SelfRegulationSCP2 (SCP2), JapaneseVowels (JV), SpokenArabicDigits (SAD), Heartbeat (HB), FaceDetection (FD), PEMS-SF (PS), and EigenWorms (EW). Notably, the PS dataset has an exceptionally high number of variables (963), while the EW dataset has extremely long time series (17984). These two datasets allow us to assess the effectiveness of our approach when dealing with large numbers of variables and long time series. We applied different image sizes according to the grid layouts for these datasets. The hyperparameter settings are

provided in Table 11, and we applied cutout [8] data augmentation methods to SCP1, SCP2, and JV datasets due to the small size of their training sets.

## B.5 Self-supervised Learning

We preliminary explored masked image modeling self-supervised pre-training on the time series line graph images. We randomly mask columns of patches with a width of 32 on each line graph within a grid cell. The masking ratio is set as 50%. We finetuned the Swin Transformer model for 10 epochs with batch size 48. The learning rate is 2e-5. Following [44], we use a linear layer to reconstruct the pixel values and employ an $\ell_1$ loss on the masked pixels:

$$\mathcal{L} = \frac{1}{\Omega(\mathbf{p}_M)} \left\| \hat{\mathbf{p}_M} - \mathbf{p}_M \right\|_1, \tag{2}$$

where $\mathbf{p}_M$ and $\hat{\mathbf{p}_M}$ are the masked and reconstructed pixels, respectively; $\Omega(\cdot)$ denotes the number of elements. With self-supervised masked image modeling, the performance improves by 1.0 in AUPRC points (percentage) from 52.8 ($\pm$ 3.8) to 53.8 ($\pm$ 3.2). The AUROC points (percentage) slightly dropped from 89.4 ($\pm$ 1.9) to 88.9 ($\pm$ 2.1).

## B.6 Full Experimental Results

We presented the full experimental results in the leave-sensors-out settings in Table 12.

Table 12: Full results in the leave-sensors-out settings on PAM dataset. The "missing ratio" denotes the ratio of masked variables.

| Missing ratio | Methods | PAM (Leave-**fixed**-sensors-out) | | | | PAM (Leave-**random**-sensors-out) | | | |
|---|---|---|---|---|---|---|---|---|---|
| | | Accuracy | Precision | Recall | F1 score | Accuracy | Precision | Recall | F1 score |
| 10% | Transformer | 60.3 ± 2.4 | 57.8 ± 9.3 | 59.8 ± 5.4 | 57.2 ± 8.0 | 60.9 ± 12.8 | 58.4 ± 18.4 | 59.1 ± 16.2 | 56.9 ± 18.9 |
| | Trans-mean | 60.4 ± 11.2 | 61.8 ± 14.9 | 60.2 ± 13.8 | 58.0 ± 15.2 | 62.4 ± 3.5 | 59.6 ± 7.2 | 63.7 ± 8.1 | 62.7 ± 6.4 |
| | GRU-D | 65.4 ± 1.7 | 72.6 ± 2.6 | 64.3 ± 5.3 | 63.6 ± 0.4 | 68.4 ± 3.7 | 74.2 ± 3.0 | 70.8 ± 4.2 | 72.0 ± 3.7 |
| | SeFT | 58.9 ± 2.3 | 62.5 ± 1.8 | 59.6 ± 2.6 | 59.6 ± 2.6 | 40.0 ± 1.9 | 40.8 ± 3.2 | 41.0 ± 0.7 | 39.9 ± 1.5 |
| | mTAND | 58.8 ± 2.7 | 59.5 ± 5.3 | 64.4 ± 2.9 | 61.8 ± 4.1 | 53.4 ± 2.0 | 54.8 ± 2.7 | 57.0 ± 1.9 | 55.9 ± 2.2 |
| | Raindrop | 77.2 ± 2.1 | 82.3 ± 1.1 | 78.4 ± 1.9 | 75.2 ± 3.1 | 76.7 ± 1.8 | 79.9 ± 1.7 | 77.9 ± 2.3 | 78.6 ± 1.8 |
| | **ViTST** | **92.8** ± 1.6 | **94.2** ± 1.3 | **93.4** ± 1.8 | **93.7** ± 1.6 | **93.1** ± 0.9 | **94.3** ± 0.9 | **94.0** ± 1.2 | **94.1** ± 1.1 |
| 20% | Transformer | 63.1 ± 7.6 | 71.1 ± 7.1 | 62.2 ± 8.2 | 63.2 ± 8.7 | 62.3 ± 11.5 | 65.9 ± 12.7 | 61.4 ± 13.9 | 61.8 ± 15.6 |
| | Trans-mean | 61.2 ± 3.0 | 74.2 ± 1.8 | 63.5 ± 4.4 | 64.1 ± 4.1 | 56.8 ± 4.1 | 59.4 ± 3.4 | 53.2 ± 3.9 | 55.3 ± 3.5 |
| | GRU-D | 64.6 ± 1.8 | 73.3 ± 3.6 | 63.5 ± 4.6 | 64.8 ± 3.6 | 64.8 ± 0.4 | 69.8 ± 0.8 | 65.8 ± 0.5 | 67.2 ± 0.0 |
| | SeFT | 35.7 ± 0.5 | 42.1 ± 4.8 | 38.1 ± 1.3 | 35.0 ± 2.2 | 34.2 ± 2.8 | 34.9 ± 5.2 | 34.6 ± 2.1 | 33.3 ± 2.7 |
| | mTAND | 33.2 ± 5.0 | 36.9 ± 3.7 | 37.7 ± 3.7 | 37.3 ± 3.4 | 45.6 ± 1.6 | 49.2 ± 2.1 | 49.0 ± 1.6 | 49.0 ± 1.0 |
| | Raindrop | 66.5 ± 4.0 | 72.0 ± 3.9 | 67.9 ± 5.8 | 65.1 ± 7.0 | 71.3 ± 2.5 | 75.8 ± 2.2 | 72.5 ± 2.0 | 73.4 ± 2.1 |
| | **ViTST** | **89.7** ± 1.7 | **91.0** ± 1.4 | **90.9** ± 1.9 | **90.8** ± 1.6 | **92.0** ± 1.4 | **93.4** ± 1.2 | **92.8** ± 1.6 | **93.0** ± 1.4 |
| 30% | Transformer | 31.6 ± 10.0 | 26.4 ± 9.7 | 24.0 ± 10.0 | 19.0 ± 12.8 | 52.0 ± 11.9 | 55.2 ± 15.3 | 50.1 ± 13.3 | 48.4 ± 18.2 |
| | Trans-mean | 42.5 ± 8.6 | 45.3 ± 9.6 | 37.0 ± 7.9 | 33.9 ± 8.2 | 65.1 ± 1.9 | 63.8 ± 1.2 | 67.9 ± 1.8 | 64.9 ± 1.7 |
| | GRU-D | 45.1 ± 2.9 | 51.7 ± 6.2 | 42.1 ± 6.6 | 47.2 ± 3.9 | 58.0 ± 2.0 | 63.2 ± 1.7 | 58.2 ± 3.1 | 59.3 ± 3.5 |
| | SeFT | 32.7 ± 2.3 | 27.9 ± 2.4 | 34.5 ± 3.0 | 28.0 ± 1.4 | 31.7 ± 1.5 | 31.0 ± 2.7 | 32.0 ± 1.2 | 28.0 ± 1.6 |
| | mTAND | 27.5 ± 4.5 | 31.2 ± 7.3 | 30.6 ± 4.0 | 30.8 ± 5.6 | 34.7 ± 5.5 | 43.4 ± 4.0 | 36.3 ± 4.7 | 39.5 ± 4.4 |
| | Raindrop | 52.4 ± 2.8 | 60.9 ± 3.8 | 51.3 ± 7.1 | 48.4 ± 1.8 | 60.3 ± 3.5 | 68.1 ± 3.1 | 60.3 ± 3.6 | 61.9 ± 3.9 |
| | **ViTST** | **86.4** ± 2.1 | **88.3** ± 1.8 | **88.0** ± 1.7 | **87.6** ± 1.7 | **88.5** ± 0.7 | **89.8** ± 0.9 | **90.1** ± 1.0 | **89.8** ± 0.9 |
| 40% | Transformer | 23.0 ± 3.5 | 7.4 ± 6.0 | 14.5 ± 2.6 | 6.9 ± 2.6 | 43.8 ± 14.0 | 44.6 ± 23.0 | 40.5 ± 15.9 | 40.2 ± 20.1 |
| | Trans-mean | 25.7 ± 2.5 | 9.1 ± 2.3 | 18.5 ± 1.4 | 9.9 ± 1.1 | 48.7 ± 2.7 | 55.8 ± 2.6 | 54.2 ± 3.0 | 55.1 ± 2.9 |
| | GRU-D | 46.4 ± 2.5 | 64.5 ± 6.8 | 42.6 ± 7.4 | 44.3 ± 7.9 | 47.7 ± 1.4 | 63.4 ± 1.6 | 44.5 ± 0.5 | 47.5 ± 0.0 |
| | SeFT | 26.3 ± 0.9 | 29.9 ± 4.5 | 27.3 ± 1.6 | 22.3 ± 1.9 | 26.8 ± 2.6 | 24.1 ± 3.4 | 28.0 ± 1.2 | 23.3 ± 3.0 |
| | mTAND | 19.4 ± 4.5 | 15.1 ± 4.4 | 20.2 ± 3.8 | 17.0 ± 3.4 | 23.7 ± 1.0 | 33.9 ± 6.5 | 26.4 ± 1.6 | 29.3 ± 1.9 |
| | Raindrop | 52.5 ± 3.7 | 53.4 ± 5.6 | 48.6 ± 1.9 | 44.7 ± 3.4 | 57.0 ± 3.1 | 65.4 ± 2.7 | 56.7 ± 3.1 | 58.9 ± 2.5 |
| | **ViTST** | **80.0** ± 2.6 | **83.7** ± 2.7 | **82.3** ± 2.4 | **81.2** ± 2.7 | **83.7** ± 1.3 | **85.5** ± 1.1 | **85.6** ± 1.4 | **85.1** ± 1.3 |
| 50% | Transformer | 21.4 ± 1.8 | 2.7 ± 0.2 | 12.5 ± 0.4 | 4.4 ± 0.3 | 43.2 ± 2.5 | 52.0 ± 2.5 | 36.9 ± 3.1 | 41.9 ± 3.2 |
| | Trans-mean | 21.3 ± 1.6 | 2.8 ± 0.4 | 12.5 ± 0.7 | 4.6 ± 0.2 | 46.4 ± 1.4 | 59.1 ± 3.2 | 43.1 ± 2.2 | 46.5 ± 3.1 |
| | GRU-D | 37.3 ± 2.7 | 29.6 ± 5.9 | 32.8 ± 4.6 | 26.6 ± 5.9 | 49.7 ± 1.2 | 52.4 ± 0.3 | 42.5 ± 1.7 | 47.5 ± 1.2 |
| | SeFT | 24.7 ± 1.7 | 15.9 ± 2.7 | 25.3 ± 2.6 | 18.2 ± 2.4 | 26.4 ± 1.4 | 23.0 ± 2.9 | 27.5 ± 0.4 | 23.5 ± 1.8 |
| | mTAND | 16.9 ± 3.1 | 12.6 ± 5.5 | 17.0 ± 1.6 | 13.9 ± 4.0 | 20.9 ± 3.1 | 35.1 ± 6.1 | 23.0 ± 3.2 | 27.7 ± 3.9 |
| | Raindrop | 46.6 ± 2.6 | 44.5 ± 2.6 | 42.4 ± 3.9 | 38.0 ± 4.0 | 47.2 ± 4.4 | 59.4 ± 3.9 | 44.8 ± 5.3 | 47.6 ± 5.2 |
| | **ViTST** | **79.7** ± 2.1 | **83.4** ± 2.3 | **81.8** ± 1.9 | **80.8** ± 2.2 | **82.8** ± 1.8 | **84.9** ± 2.0 | **84.9** ± 1.8 | **84.4** ± 1.9 |

