# OpenReview forum: "Time Series as Images: Vision Transformer for Irregularly Sampled Time Series"
_NeurIPS.cc/2023/Conference — NeurIPS 2023 poster_

### Official Review · Reviewer_BBUf · 2023-06-30

**Soundness:** 3 good
**Presentation:** 3 good
**Contribution:** 3 good
**Rating:** 3
**Confidence:** 4

**Summary:**

This paper explores a very interesting direction to represent time series as plotted images and then stack vision transformers to perform representation learning. Following this idea, this paper has conducted empirical studies on some classification benchmarks of both irregularly sampled time series and regularly sampled ones. By leveraging this new data representation, the proposed method has obtained the state-of-the-art performance on three irregular time-series benchmarks and also produced competitive results on multiple regular time-series dataests.

**Strengths:**

In my view, the biggest strength of this paper is the first successful demonstration of applying image-driven methods on representation learning of irregularly sampled time series. While after reading the related work section, I have also learned that this paper is not the first to represent time series as real images in general. Neverthess, I think the successful demonstration of image-based time-series representation learning on time-series classification tasks is interesting and has sufficient novelty.

**Weaknesses:**

However, I do have several concerns on the experimental results at the current stage.

- Given the relatively high performance variations when employing different plotting configurations and specific early stopping epochs on different datasets, I would like to know the detailed hyper-parameter tuning procedure when you develop ViTST. Are you tuning these options directly based on the feedback from the test set? If not, can you elaborate on how to divide the validation part and please report the details of performance on both validation and test sets if possible.
- This paper includes some ablation tests, but I think the existing results are far from enough to convince me about the robustness of ViTST. For example, as table 3 shows, this method is sensitive to the existences of interpolation, markers, colors. I want to know that whether ViTST is robust to 1) different line styles and widths given the existence of interpolation, 2) different marker sizes and types given the existence of markers, and 3) different color permutations assigned to multivariate time series. My central confusions lie in which aspects play indispenable roles in making ViTST work and whether ViTST is robust to those irrelevant options. Moreover, I would like to see more ablation results about grid layouts and image sizes. Tables 2, 3, 4 only cover a few, and there seem to large performance variations, too.
- Another question bothering me is that why ViTST can obtain the best performance for irregularly sampled time series but is only roughly comparable to existing solutions for regular time series.
- I have tried the code but cannot reproduce the results.

**Questions:**

See the weakness part.

**Limitations:**

The major limitation is that we are not clear many other critical factors ensuring the success of ViTST and whether it is robust to different plotting setups.

---

> ### Author Rebuttal · Authors · 2023-08-09
>
> We would like to express our sincere gratitude for the valuable feedback and constructive comments. Please find our response addressing the concerns below.
>
> **Response to W1: Detailed hyper-parameter tuning procedure and Robustness test**
>
> The training/validation/test sets are randomly split. Initially, our selections for line style, line width, marker size, and colors were based on heuristic methods, with the goal of clearly illustrating patterns in human observation (on the training set). Following this, we adjusted the hyper-parameters based on the performance observed on the validation set.
> The performance of the validation and test set along training process on the P12 dataset is as follows:
> | Steps | Training loss | Validation AUROC | Validation AUPRC | Test AUROC | Test AUPRC |
> | --- | --- | --- | --- | --- | --- |
> | 40 | 0.35 | 0.6988 | 0.269 | 0.6502 | 0.2098 |
> | 80 | 0.5951 | 0.7843 | 0.373 | 0.7953 | 0.3698 |
> | 120 | 0.5066 | 0.8124 | 0.3915 | 0.8286 | 0.4113 |
> | 160 | 0.4837 | 0.8252 | 0.4091 | 0.8409 | 0.4686 |
> | 200 | 0.4847 | 0.8304 | 0.4471 | 0.8531| 0.4857 |
> | 240 | 0.4395 | 0.8336 | 0.4569 | 0.8574 | 0.4781 |
> | 280 | 0.4151 | **0.8357** | 0.4502 | 0.8578 | 0.4881 |
> | 320 | 0.4017 | 0.8344 | 0.4443 | 0.8558 | 0.4882 |
> ...
>
> We choose the checkpoint that achieves the best AUROC score on the validation set for testing.
>
> The robustness study on choices of image creations are shown below:
>
> **Color:**
> We manually assigned colors, with the principle of making adjacent line graphs have distinguishable colors. For comparison, we randomly selected colors over 3 times. The results are provided below.
> Our manual choices weren't optimal for the P19 dataset; random color selections yielded superior results, boosting the AUPRC score to 54.5. The PAM dataset, however, remained less affected by changes in line color.
>
> | P19 | AUROC | AUPRC |
> | --- | --- | --- |
> | Default | 89.4$\pm$1.9 | 52.8$\pm$3.8 |
> | Random color 1 | 89.3$\pm$1.4 | 53.6$\pm$1.9 |
> | Random color 2 | 89.1$\pm$1.9 | 54.5$\pm$3.5 |
> | Random color 3 | 88.9$\pm$2.1 | 52.9$\pm$2.7 |
>
> | P12 | AUROC | AUPRC |
> | --- | --- | --- |
> | Default | 85.6$\pm$1.1 | 49.8$\pm$2.5 |
> | Random color 1 | 85.0$\pm$1.6 | 49.3$\pm$3.7 |
> | Random color 2 | 83.9$\pm$1.5 | 48.6$\pm$2.7 |
> | Random color 3 | 85.2$\pm$1.9 | 49.2$\pm$3.2 |
>
> | PAM | Accuracy | Precision | Recall | F1 Score |
> | --- | --- | --- | --- | --- |
> | Default | 96.1$\pm$0.7 | 96.8$\pm$1.1 | 96.5$\pm$0.7 | 96.6$\pm$0.9 |
> | Random color 1 | 94.5$\pm$1.3 | 95.8$\pm$0.8 | 95.0$\pm$1.2 | 95.3$\pm$1.0 |
> | Random color 2 | 95.2$\pm$1.5 | 96.5$\pm$1.1 | 95.2$\pm$1.4 | 95.8$\pm$1.3 |
> | Random color 3 | 95.4$\pm$1.7 | 96.6$\pm$0.9 | 95.9$\pm$1.5 | 95.9$\pm$1.1 |
> | Random order 1 | 95.5$\pm$1.0 | 96.8$\pm$0.8 | 95.7$\pm$1.0 | 96.2$\pm$0.9 |
> | Random order 2 | 95.2$\pm$0.7 | 96.6$\pm$0.6 | 95.6$\pm$0.8 | 96.0$\pm$0.7 |
> | Random order 3 | 95.4$\pm$0.8 | 96.6$\pm$0.7 | 95.8$\pm$1.1 | 96.1$\pm$0.7 |
>
> **Line width/style and marker size/style:**
> We configured line style to solid, and marker as '*'. For the P12 and P19 datasets, the line width and the marker size are set to 1 and 2, while for the PAM dataset, they are set to 0.5 and 0.5. The guiding principle is to optimize the line graphs for human visualization, ensuring that the lines effectively show trends and markers clearly indicate observed data without obscuring the lines. We also tested the other configuration. Due to limited space, we only show the results on P19 here.
>
> | Line style | Line width | Marker | Marker size | AUROC | AUPRC |
> | --- | --- | --- | --- | --- | --- |
> | Solid | 1 | * | 2 | 89.4$\pm$1.9 | 52.8$\pm$3.8 |
> | Dotted | 1 | * | 2 | 88.7$\pm$2.0 | 53.3$\pm$3.4 |
> | Dashed | 1 | * | 2 | 88.5$\pm$2.8 | 53.3$\pm$4.2 |
> | Solid | 2 | * | 2 | 88.5$\pm$2.3 | 53.6$\pm$3.4 |
> | Solid | 3 | * | 2 | 88.9$\pm$1.9 | 52.8$\pm$3.2 |
> | Solid | 1 | o | 2 | 88.8$\pm$2.6 | 53.2$\pm$3.8 |
> | Solid | 1 | ^ | 2 | 88.8$\pm$2.2 | 53.1$\pm$4.0 |
> | Solid | 1 | * | 1 | 88.6$\pm$2.4 | 52.6$\pm$3.6 |
> | Solid | 1 | * | 3 | 89.2$\pm$2.1 | 53.1$\pm$4.2 |
>
> From the result, we can see that the line style has a significant impact on the performance, whereas markers appear to have minimal impact. This observation is consistent with our initial heuristic that a solid line would likely be more effective, and markers might not make big influence. As for other factors, an optimal combination of line and marker could potentially enhance visualization, thereby improving performance. We did not conduct comprehensive hyperparameter tuning for these variables. As such, it seems that the peak performance on P19 has not yet been fully explored.
>
> **Response to Q: why ViTST can obtain the best performance for irregularly sampled time series but is only roughly comparable to existing solutions for regular time series?**
> As shown in Table 4, our ViTST's average accuracy is only behind TST, which is a transformer that directly operates on numerical time series. Given the shared architecture between ViTST and TST, we assume their performance gap might be attributed to the fact that extracting patterns from complete fixed-sized numerical time series data is more direct and introduces less noise than operating on the image data transformed from time series.  However, when there is missing data and imputation is required to convert the data into fixed-sized input can introduce much noise, leading to suboptimal performance of standard numerical-based specialized methods. However, our method might not be such sensitive to the missing part in the line graph. By simple linear interpolation, the model might be able to approximate the pattern and trend from the line graph, ensuring that ViTST remains effective even with irregularly sampled data.

---

> > ### Comment · Reviewer_BBUf · 2023-08-17
> > **Thanks for the reply**
> >
> > I think this paper is interesting so I try to reproduce your results with your code.
> >
> > However, I find some crucial issues and would like to hear your answers.
> >
> > - When deciding how to plot time-series images, especially the y-axis ranges, your code seems to use all data (including both train and test), which is improper because this step is just like doing normalization for time series, you should only use the training data to determine these hyper-parameters.
> > - It is hard for me to reproduce your results reported in the paper. I mainly follow your provided code to run the experiments. Here is what I have obtained on P12 using ViT.
> >
> > Accuracy      = 71.8 +/- 9.3
> > AUPRC         = 16.1 +/- 1.2
> > AUROC         = 54.0 +/- 2.1
> > Precision     = 20.0 +/- 9.0
> > Recall        = 22.6 +/- 13.6
> >
> > Would you like to provide detailed instructions for me to replicate your results? I think tuning such a vision model must be very tricky. So, I change the score to 'Reject' unless I can reproduce the results.

---

> > > ### Author Response · Authors · 2023-08-20
> > > **Response to the reviewer's reply**
> > >
> > > Thank you for your interest in our work. We value your feedback and would like to address your concerns as follows:
> > >
> > > Our setup resembles the scenario where the possible value range of variables is predetermined or well-understood. In cases where it isn't, the range can be derived when the test set features (not labels) are available in practice. Furthermore, with an ample amount of training data, the disparity between the distribution of the training set and the entire dataset is marginal., when there's a sufficient volume of training data, the disparity between the distribution of the training set and the actual distribution (represented by the whole dataset in the experiments) is minimal.
> > >
> > > Taking your feedback into account, we also explored rescaling based on the training set's distribution. The results are as follows:
> > >
> > > | P19 | AUROC | AUPRC |
> > > | --- | --- | --- |
> > > | ViTST | 89.2$\pm$2.0 | 53.1$\pm$3.4 |
> > >
> > > | P12 | AUROC | AUPRC |
> > > | --- | --- | --- |
> > > | ViTST | 84.2$\pm$1.7 | 48.0$\pm$4.7 |
> > >
> > > | PAM | Accuracy | Precision | Recall | F1 Score |
> > > | --- | --- | --- | --- | --- |
> > > | ViTST | 93.5$\pm$1.4 | 94.9$\pm$1.2 | 94.1$\pm$1.3 | 94.4$\pm$1.2 |
> > >
> > > Our approach still achieves superior performance. On the extensive dataset P19, the performance remains consistent, underscoring that as training data grows, the performance difference between the two settings becomes negligible.
> > >
> > >
> > > To reproduce the experimental results, you can follow the instruction below:
> > > First, ensure that you've acquired the processed data as directed in the `README` of our code repository.
> > > To create the images, execute the following commands sequentially:
> > > ```code
> > > cd dataset/P12data/process_scripts
> > > python ParamDescription.py
> > > python ConstructImage.py
> > > ```
> > > Upon completion, you should have a directory named `differ_interpolation_6*6_images` created under `dataset/P12data/processed_data/`, containing all the created images .
> > >
> > > For training, navigate to the `code/Vision-Text/` directory. We provided a script for the P19 dataset in the README. For the P12 dataset, you can use the following script:
> > > ```code
> > > CUDA_VISIBLE_DEVICES=0 python3 run_VisionTextCLS.py \
> > >     --image_model swin \
> > >     --text_model roberta \
> > >     --freeze_vision_model False \
> > >     --freeze_text_model False \
> > >     --dataset P12 \
> > >     --dataset_prefix differ_interpolation_6*6_ \
> > >     --seed 1799 \
> > >     --save_total_limit 1 \
> > >     --train_batch_size 48 \
> > >     --eval_batch_size 196 \
> > >     --logging_steps 20 \
> > >     --save_steps 100 \
> > >     --epochs 4 \
> > >     --learning_rate 2e-5 \
> > >     --n_runs 1 \
> > >     --n_splits 5 \
> > >     --cutout_num 16 \
> > >     --cutout_size 16 \
> > >     --do_train \
> > > ```
> > > This script uses the default vision model Swin. If you're inclined to use the ViT model, simply modify the ``--image_model`` argument to `vit`.
> > >
> > > If you still encounter challenges in reproducing the results using the provided instructions, please provide details on the specific parameters and procedures you implemented. It seems that your current results resemble those of a ViT model trained from scratch, knowing the detailed parameter and argument you implemented will aid us in identifying any inconsistencies and offering more accurate guidance.

---

### Official Review · Reviewer_kdpF · 2023-07-03

**Soundness:** 2 fair
**Presentation:** 3 good
**Contribution:** 2 fair
**Rating:** 5
**Confidence:** 5

**Summary:**

The paper investigates the use of pre-trained Vision Transformers (ViT) for both regularly and irregularly sampled time series classification. Two primary aspects are discussed: Transforming time series into images and performing time series classification using pre-trained ViT. Through a series of experiments, the authors illustrate the feasibility of this research direction.

**Strengths:**

The paper is well-organized and presents an engaging perspective on time series classification, particularly focusing on the potential of ViT in relevant tasks. The authors conducted extensive empirical evaluations to showcase the effectiveness of the proposed framework, which is a strong aspect of the work.

**Weaknesses:**

+ The paper falls short in providing an in-depth analysis of several critical aspects, such as the advantages of using ViT over other foundational models for time series, factors contributing to successful cross-domain transfer when employing ViT for time series classification, and how key design factors impact performance. See the listed questions for details.
+ In addition, some crucial technical details are missing. For example, it is unclear which layers in ViT are frozen during downstream fine-tuning. Justification for the use of line graphs over other visualization strategies (e.g., frequency maps) is not adequately discussed in Section 3.1. The model's efficiency compared to specialized models in time series modeling is also not clear.
+ There are some writing issues. For example, the title should be “…for Irregularly Sampled Time Series Classification” unless the effectiveness of ViTST is evident on other time series tasks such as forecasting and imputation. Besides, it is unknown whether the backbone ViT model are frozen in Fig. 2.

**Questions:**

+ What distinguishes the use of ViT for time series modeling from other foundational models like language or acoustic models (e.g., Voice2Series)? Are there unique advantages, beyond performance, to using ViT?
+ What theoretical factors contribute to such a successful cross-domain transfer? According to the experiments, some factors like line colors may significantly affect the model performance. A follow-up question is how to claim the robustness of the proposed framework in general time series classification when changing line shapes, colors, or even time series datasets to other domains other than healthcare.
+ What types of interpolation are used in the experiments?
+ What is the motivation behind Section 4.5? Is it for a second pre-training based on pre-trained ViT or is it for pre-training from scratch?
+ In the transformation of time series to images, why do the authors opt to combine different univariate time series into a single image, as opposed to potential alternative implementations like introducing an additional layer to fuse different input images before feeding them into ViT?

**Limitations:**

The authors have not adequately addressed the limitations and potential negative societal impacts of their work.

---

> ### Author Rebuttal · Authors · 2023-08-09
>
> We appreciate your valuable feedback. Here is our response to address your concerns:
>
> **Response to weakness**:
>  - **Layer frozen**: In our implementation, we did not freeze any of the layers. All the parameters are tunable during fine-tuning on the time series dataset.
>  - **Visualization strategy**: Our primary is to showcase the potential of a simple and intuitive approach that uses vision transformers for time series modeling. Line graphs were selected as they are straightforward for humans to interpret. We show that such a strategy has already achieved competitive results. Our current work serves as a pivotal first step to exploring the adaptation of pre-trained vision transformers for time series modeling, and we demonstrate its feasibility. We value the potential of exploring other visualization techniques to further enhance the approach in future work.
>  - **Model Efficiency**: we present the performance comparison of our method with several representative specialized methods on the irregularly sampled time series dataset in Table 2 and Figure 3. We also tested on ten more diverse time series classification datasets as described in Section 4.4. The results from these experiments indicate the efficacy and versatility of our approach in addressing a diverse range of time series shapes and types.
>  - **Writting issues**: Thank you for pointing it out. We will change the title as suggested. As for Figure 2, none of the layers or parameters are frozen during fine-tuning on the time series dataset. We will add a notification for clarity.
>
> **Response to questions**:
>  - **Comparison with other fundamental methods for time series modeling**: Our work aims to showcase that vision transformers can be adapted for time series modeling. We do not claim it is the optimal or only foundation model that could work for this purpose. However, compared with language models and voice models (Voice2Series) for time series modeling, our vision-based approach might offer the following advantages beyond performance: (1) **Simplicity and Compression**: Translating time series data into images provides a more compressed and straightforward representation than converting them into language. In our preliminary experiments, we attempted to transform time series data into text. We observed challenges, especially with multivariate time series that have a significant number of variables or long sequences. This is because most of affordable language models have a context window ranging less than 4096 tokens. Even if each observation is considered a single token, accommodating time series that surpasses this context window size is problematic. In contrast, time series data of any dimension can seamlessly fit into a single image, unconstrained by length or magnitude. (2) **Generality**: While Voice2Series showcases potential in handling univariate time series, its applicability for multivariate and irregularly sampled time series remains unclear. (3) **Maturity of Domain**: using our vision-based method provides the opportunity to leverage well-studied techniques in the established computer vision field.
>  - **Theoretical factors contribute to successful transfer**: The success is based on the hypothesis that the model pre-trained on large image datasets can capture generic features within the images, which can then be fine-tuned for specific tasks like time series classification. Our experiments indicate that certain factors within the images, such as line colors/styles/width and grid orders (more details can be found [here](https://openreview.net/forum?id=ZmeAoWQqe0&noteId=13G2qHdC3K)), and also the pre-training of vision transformers do influence performance. We recognize the need for a more comprehensive theoretical analysis in future studies. This work represents the initial exploration of utilizing vision transformers for time series modeling, and our results demonstrate its feasibility and promise.
> - **Robustness to general time series classification tasks**: we conducted experiments on two healthcare datasets (P19/P12), a human activity dataset PAM, and also ten general time series classification datasets as outlined in Section 4.4. In total, we assessed our method on 13 distinct datasets, employing distinguished image rendering settings for each dataset. Our results show that as long as the images are rendered with the same setting in the training set and evaluation set, the performance of our approach is consistently competitive.
>  - **Interpolation**: Taking the minimalist approach, our approach employs linear interpolation, i.e., using straight lines to connect consecutive data points on the line graph. The main focus of our work is to show that adapting vision transformers for modeling time series line graphs work, and our current results have proven that. However, we value the potential of exploring other interpolation methods to further enhance the performance.
>  - **Motivation behind Section 4.5**: As mentioned above, our approach bridges the time series and computer vision domain, opening up the possibility of leveraging well-studied computer vision techniques into the time series domain. Therefore, in Section 4.5, we explored the prevalent masked image modeling method for time series modeling. Likewise, we started the self-supervised learning with pre-trained checkpoints.
>  - **Motivation of image combination**: In our approach, we plot the time series of each variable, which can be seen as a sub-image. These sub-images are assembled into a single "super image.". In our preliminary experiment, we tested obtaining separate image representations for each sub-image and concatenating and feeding them into a prediction layer. However, this approach consistently underperformed the unified image processing technique. We hypothesize that integrating them into a single image enables the model to discern more intricate correlations among different variables directly from raw features.

---

> > ### Comment · Area_Chair_zrLs · 2023-08-18
> > **Response to author rebuttal**
> >
> > Dear reviewer,
> >
> > The author rebuttal appears to have presented several targeted responses to your questions.
> >
> > Are your questions appropriately addressed?
> > If they are, would you consider re-assessing your score in light of them.
> > If not, please do provide additional context and feedback to the author.
> >
> > In either case, please provide an acknowledgement of the effort the authors put in, why your questions have (or have not) been addressed and what your assessment of the work is in light of this evidence with a view to reach consensus with the other reviewers on this work.
> >
> > -AC

---

> > ### Comment · Reviewer_kdpF · 2023-08-18
> > **Reviewer's reply to the rebuttal**
> >
> > I would like to express my gratitude to the authors for their detailed responses, which have helped to clarify several of the issues that were previously raised. After carefully reviewing the manuscript, considering the comments of other reviewers, and evaluating the authors' rebuttals, I am inclined to adjust my score to borderline reject. However, I must emphasize that the paper, in its current form, is not yet in a condition suitable for publication at a conference of NeurIPS' caliber. There are several aspects that require more comprehensive analysis, and these issues have not been fully resolved in the authors' rebuttal:
> >
> > + What theoretical factors contribute to such a successful cross-domain transfer? As this is the main argument in this research, answering this question is crutial.
> > + I see no experiements related to model efficiency comparisons. The aforementioned results (e.g., Tab. 2 and Fig. 3) are not related to this. Otherwise, it makes no sense to claim the *simplicity and compression* in this research.
> > + The authors' rebuttal does not convincingly justify the impact of line shapes and colors on the robustness of the proposed method.
> > + Regarding the motivation of image combinbation, I see no results and discussion related to the mentioned preliminary experiemnts. Please consider to clarify this: *In our preliminary experiment, we tested obtaining separate image representations for each sub-image and concatenating and feeding them into a prediction layer. However, this approach consistently underperformed the unified image processing technique.*

---

> > > ### Author Response · Authors · 2023-08-20
> > > **Response to the reviewer's reply**
> > >
> > > Thank you for your thoughtful feedback. We appreciate the opportunity to clarify and address the concerns raised.
> > >
> > > **R1:** The success in cross-domain transfer, as we hypothesized, might be rooted in the pre-training stage with masked image modeling. As found in [1], vision transformer models pretrained with masked image modeling introduce locality inductive bias across all layers, enlarging the receptive field. These models have shown superior performance in tasks with weak semantics like geometric and motion or fine-grained classification, which is analogous to our task. Nonetheless, we acknowledge the importance for more in-depth explorations and intend to pursue this in future work.
> > >
> > > **R2:** Regarding **Simplicity and compression**, our comparison of simplicity and compression is with **language foundational models**, in response to the reviewer's question on "**What distinguishes the use of ViT for time series modeling from other foundational models like language or acoustic models (e.g., Voice2Series)?**". As we mentioned, "Translating time series data into images provides a more compressed and straightforward representation **than converting them into language**.  In our preliminary experiments, we attempted to transform time series data into text. We observed challenges, especially with multivariate time series that have a significant number of variables or long sequences. This is because most affordable language models have a context window ranging less than **4096** tokens. Even if each observation is considered a single token, accommodating time series that usually has tens of thousands of observations is problematic. In contrast, time series data of any dimension can seamlessly fit into a single image, unconstrained by length or magnitude. " As for the efficiency compared to other models, we do not claim our model is more efficient than other non-vision baselines. The results and discussion can be found at https://openreview.net/forum?id=ZmeAoWQqe0&noteId=kAotjOzWij.
> > >
> > > **R3:** We apologize for previously omitting the link to the results on the influence of line shapes and colors on our method's robustness. Kindly see https://openreview.net/forum?id=ZmeAoWQqe0&noteId=13G2qHdC3K for the results and discussion.
> > >
> > > **R4**: In our preliminary experiments, we tested obtaining the patch representations in each sub-image separately and then concatenated the patch representations of all sub-images to feed into the final prediction layer for prediction. Compared with our default combination approach, the difference lies in whether a patch can attend to patches from other sub-images and the position embedding when learning the patch/image representations. All the other parameters remain the same. We tested on ViT and the performance comparisons are shown below:
> > >
> > > | P19 | AUROC | AUPRC |
> > > | --- | --- | --- |
> > > | ViT | 87.9$\pm$2.5 | 51.6$\pm$3.7 |
> > > | ViT-Subimage| 85.1$\pm$1.5 | 47.9$\pm$3.4 |
> > >
> > > | P12 | AUROC | AUPRC |
> > > | --- | --- | --- |
> > > | ViT | 84.8$\pm$1.3 | 48.1$\pm$3.8 |
> > > | ViT-Subimage| 77.6$\pm$3.2 | 35.5$\pm$6.2 |
> > >
> > > | PAM | Accuracy | Precision | Recall | F1 score |
> > > | --- | --- | --- | --- | --- |
> > > | ViT | 93.4$\pm$0.7 | 94.7$\pm$0.9 | 94.1$\pm$0.7 | 94.3$\pm$0.7 |
> > > | ViT-Subimage| 90.4$\pm$1.3 | 92.9$\pm$1.0 | 91.12$\pm$0.9 | 91.9$\pm$1.0 |
> > >
> > > It is evidenced that attending to patches from other sub-images offers advantages. This likely allows for capturing cross-variable correlations at a granular level within the self-attention layers, as opposed to only at the final linear prediction layer.
> > >
> > > **References:**
> > >
> > > [1] Xie, Zhenda, et al. "Revealing the dark secrets of masked image modeling." Proceedings of the IEEE/CVF Conference on Computer Vision and Pattern Recognition. 2023.

---

> > > > ### Comment · Reviewer_kdpF · 2023-08-21
> > > > **Reviewer's reply to the rebuttal**
> > > >
> > > > Thank you to the authors for their comprehensive response, which has addressed the majority of my concerns. I strongly recommend incorporating the discussions and findings from R2, R3, and R4 into the manuscript to elevate its quality. While R1 remains unresolved, I am confident that the authors recognize the areas to enhance in presenting their research. Given these revisions, I am inclined to adjust my evaluation to a borderline accept.

---

> > > > > ### Author Response · Authors · 2023-08-21
> > > > > **Response to the reviewer's reply**
> > > > >
> > > > > Thank you for your insightful feedback and suggestions. We will integrate the discussions and findings from R2, R3, and R4 into our paper's updated version. Additionally, R1 will be included in our future work section. Thank you again!

---

### Official Review · Reviewer_iUCn · 2023-07-10

**Soundness:** 3 good
**Presentation:** 3 good
**Contribution:** 2 fair
**Rating:** 4
**Confidence:** 5

**Summary:**

The paper focuses on learning from irregularly sampled time series data. The paper presents a simple approach that converts irregularly sampled time series into an image where different input channels are line graphs. The converted images are then modeled using a standard Transformer model. Experiments show that the proposed approach outperforms the SOTA approaches on several datasets.


**Strengths:**

- The paper focuses on the task of learning from irregularly sampled data which is important in many domains.
- Experimental results show the effectiveness of the approach when compared to baselines and other recent approaches.
- The paper is clear and well written.


**Weaknesses:**

- The irregularly sampled time series data are represented by a line graph where observed points are connected using a straight line which is a very adhoc way to deal with missing values and irregularity.
- The novelty is marginal as the proposed approach mainly uses a standard Transformer model where time series data is fed as images.
- The standard deviation of results in Table 2 seems very high. It is not immediately clear if the proposed approach achieves statistically significant performance or it's just noise.
- The proposed approaches uses static features present with the data. It is not immediately clear how these static features were used with the baseline approaches.
- The paper notes that all the methods were trained for 20 epochs. Did they all converge in 20 epochs? The baseline approach should be trained till convergence.
- The paper is missing experiments on MIMIC-III dataset, most commonly used dataset in this space.
- The paper is missing comparisons with recent ODE based approaches which achieve SOTA performance.
- Does the size of image change if the sequence length changes?

**Questions:**

I have mentioned my concerns in the weaknesses section.

**Limitations:**

No.

---

> ### Author Rebuttal · Authors · 2023-08-09
>
> We sincerely appreciate your valuable feedback. Below is our response addressing your concerns.
>
> **Response to W1: adhoc way to deal with missing values and irregularity**
>
> While our method may seem ad-hoc, it is simple and notably effective. It largely simplifies model design for irregular time series modeling and bridges time series analysis with the computer vision domain. As an initial exploration, we demonstrate such an approach is feasible and promising. Further exploration of better way to handle missing data is encouraged.
>
> **Response to W2: marginal novelty as it mainly uses a standard Transformer model**
>
> Our paper primarily demonstrates that the pre-trained vision transformers, prevalent in computer vision tasks, can be easily adapted for time series classification by representing these series as images. Our approach is simple, effective, and general. It not only largely simplifies dedicated model design for irregular time series modeling but also bridges the time series domain with the computer vision domain.
> We recognize the potential of developing vision transformers tailored for time series modeling as a promising direction for future improvement. However, our current work represents the critical first step and demonstrates the feasibility and promise of this approach.
>
> **Response to W3: standard deviation of results in Table 2**
>
> As shown in Table 2, our approach exhibits the lowest standard deviation among all compared methods, except for Transformer-mean on the P19 dataset.
>
> **Response to W4: how the static feature is used in the baseline approaches**
>
> The static feature is converted into a vector and incorporated with the time series data for modeling in the baseline approaches. In our experiment, we convert the static feature into natural language sentences and use Roberta to decode it.
>
> As mentioned in B2 in the Appendix, we have also tried to convert the static feature into a vector and utilize MLP to encode it. Specifically, we used a 2-layer MLP with a hidden dimension of 128 and an output dimension of 96 to encode the static feature. The obtained output is concatenated with the representation learned by the vision transformer to make the prediction. We also presented the performance of our approach without using static features (only a Swin to model time series image data). The corresponding results for these experiments are detailed below.
>
> | P19 | AUROC | AUPRC |
> | --- | --- | --- |
> | Swin | 89.4$\pm$1.8 | 50.2$\pm$3.0 |
> | Swin-MLP | 88.6$\pm$1.3 | 51.4 $\pm$3.7 |
> | Swin-Roberta | 89.4$\pm$1.9 | 52.8$\pm$3.8 |
>
> | P12 | AUROC | AUPRC |
> | --- | --- | --- |
> | Swin | 84.3$\pm$0.6 | 49.3$\pm$3.7 |
> | Swin-MLP | 84.6$\pm$0.9 | 48.7 $\pm$3.2 |
> | Swin-Roberta | 85.6$\pm$1.1 | 49.8$\pm$2.5 |
>
> As can be seen, when both our method and the baselines use the vector converted from static feature for modeling, our approach still outperforms the baselines. Utilizing a language model to encode patient information, which preserves its inherent meaning, appears more effective than simply employing an MLP to process vectorized features. Furthermore, leveraging a language model may offer broader applicability, especially in contexts with textual data like clinical notes.
>
> **Response to W5: did all the baseline coverage in 20 epochs**
> All baseline models were trained for a maximum of 20 epochs and converged within this period. This setup is consistent with previous work [1].
>
> **Response to  W6/7: missing experiment on MIMIC-III and comparison with recent ODE-based methods**
> We will endeavor to incorporate experiments on MIMIC-III and comparisons with recent ODE-based methods in our revised version.
>
> **Response to W8: Does the size of image change if the sequence length changes?**
>
> No, the sequence length will not directly influence the size of images. Regardless of the length, any time series can be fitted into a line graph in a grid cell. The dimensions of this grid, which are user-defined, will determine the overall image size. To illustrate, let's consider the EW dataset detailed in Section 4.4. It has a sequence length of 17984 with 6 variables. This can be represented in a 2x3 grid. We choose the grid size of 128x128, resulting in an image size of 256x384. However, one can also choose any other grid size and subsequently change the corresponding image size.
>
> [1] Zhang, Xiang, et al. "Graph-guided network for irregularly sampled multivariate time series." arXiv preprint arXiv:2110.05357 (2021).

---

> > ### Comment · Area_Chair_zrLs · 2023-08-18
> > **Response to author rebuttal**
> >
> > Dear reviewer,
> >
> > The author rebuttal appears to have presented several targeted responses to your questions.
> >
> > Are your questions appropriately addressed?
> > If they are, would you consider re-assessing your score in light of them.
> > If not, please do provide additional context and feedback to the author.
> >
> > In either case, please provide an acknowledgement of the effort the authors put in, why your questions have (or have not) been addressed and what your assessment of the work is in light of this evidence with a view to reach consensus with the other reviewers on this work.
> >
> > -AC

---

### Official Review · Reviewer_GiK3 · 2023-07-24

**Soundness:** 3 good
**Presentation:** 4 excellent
**Contribution:** 3 good
**Rating:** 7
**Confidence:** 5

**Summary:**

This paper describes a surprisingly simple and effective approach for applying computer vision Transformer models to time series classification. Multivariate time series inputs are plotted in a grid to produce images that are used to fine-tune pretrained vision Transformer models. This approach achieves state-of-the-art results on irregularly sampled datasets and competitive results on regularly sampled datasets. Aside from the empirical efficacy, the results are relevant to understanding the capabilities of Transformers and the effectiveness of cross-domain pretraining.

**Strengths:**

I think the basic finding of the paper - that pre-trained vision Transformers are effective at performing time series classification - is original and significant. It raises very interesting questions about the general efficacy of pre-trained Transformers and the relationship between time series and visual data that could inform future work. In a practical sense, I have some concerns detailed below but I generally appreciate the simplicity and efficacy of the proposed approach.

The basic results are convincing and the ablation results are extensive and informative. Most of the questions I would have had about the model are already addressed in the main paper or Appendix.

The paper was well-written and clear. Results are reproducible since full code was provided.

**Weaknesses:**

To use the proposed method requires making a number of somewhat arbitrary decisions on how to convert time series to images: what layout to use, what order to put the time series in, what colours to use, line width/style, etc. The results in Table 3 and the Appendix show that decisions like these can affect performance even though they don't strictly change the information contained in the input. While these factors can be empirically tuned, the lack of robustness to seemingly arbitrary choices is concerning.

In particular, the use of matplotlib to generate inputs makes the input construction somewhat poorly controlled. While I appreciate the simplicity of using an existing library, this leaves many implementation details about how exactly plots are rendered up to the library. Again, given that purely visual variations can affect model performance, it's not clear how important these implementation details are. I would have preferred to see a more precise approach that formally determined what numeric value is assigned to each pixel, making these details apparent.

The range of irregularly sampled time series datasets is somewhat small to draw general conclusions from, but this is in line with existing work in this area.

**Questions:**

How were line colours selected? If random, what distribution of colours was used? Is there any justification for the specific choice made?

For your self-supervised learning experiments, did you start with a pretrained model?

**Limitations:**

Limitations are not explicitly discussed. No societal impact concerns come to mind, but I think technical limitations should have been addressed. These could include comparing training and inference resource requirements to other models and acknowledging points like those I raised in the Weaknesses section if the authors agree they are limitations.

---

> ### Author Rebuttal · Authors · 2023-08-09
>
> We sincerely appreciate the reviewer's recognition of the value and strengths of our work, as well as the constructive feedback and suggestions. We aim to address your concerns in our response as follows:
>
> **Response to W1: Robustness to seemingly arbitrary choices**
>
> Our approach does involve several decisions during the image transformation. We have conducted ablation studies on the choice of layout. While we have already conducted ablation studies concerning the layout choice, we've further expanded our investigation to cover additional parameters.
>
> **Order and Color:**
> Our default approach orders the variables based on the number of observations within each. For comparison, we shuffled this order three times and assessed the performance. We manually assigned colors, with the principle of making adjacent line graphs have distinguishable colors. Likewise, we randomly selected colors over 3 times for comparison. The results are provided below.
>
> On the P19 and P12 datasets which have varying observations across different variables, the sorted order outperforms randomly shuffled orders. On the PAM dataset, where each variable has a consistent number of observations, the order doesn't have a significant impact.
> Regarding color selection, our manual choices weren't optimal for the P19 dataset; random color selections yielded superior results, boosting the AUPRC score to 54.5. The PAM dataset, however, remained less affected by changes in line color.
>
> | P19 | AUROC | AUPRC |
> | --- | --- | --- |
> | Default | 89.4$\pm$1.9 | 52.8$\pm$3.8 |
> | Random order 1 | 88.3$\pm$1.8 | 49.9$\pm$3.2 |
> | Random order 2 | 88.2$\pm$1.6 | 51.0$\pm$3.9 |
> | Random order 3 | 88.5$\pm$2.3 | 51.9$\pm$2.6 |
> | Random color 1 | 89.3$\pm$1.4 | 53.6$\pm$1.9 |
> | Random color 2 | 89.1$\pm$1.9 | 54.5$\pm$3.5 |
> | Random color 3 | 88.9$\pm$2.1 | 52.9$\pm$2.7 |
>
> | P12 | AUROC | AUPRC |
> | --- | --- | --- |
> | Default | 85.6$\pm$1.1 | 49.8$\pm$2.5 |
> | Random order 1 | 84.3$\pm$2.2 | 48.0$\pm$4.5 |
> | Random order 2 | 84.2$\pm$1.7 | 47.6$\pm$3.7 |
> | Random order 3 | 83.9$\pm$1.5 | 46.9$\pm$4.0 |
> | Random color 1 | 85.0$\pm$1.6 | 49.3$\pm$3.7 |
> | Random color 2 | 83.9$\pm$1.5 | 48.6$\pm$2.7 |
> | Random color 3 | 85.2$\pm$1.9 | 49.2$\pm$3.2 |
>
> | PAM | Accuracy | Precision | Recall | F1 Score |
> | --- | --- | --- | --- | --- |
> | Default | 96.1$\pm$0.7 | 96.8$\pm$1.1 | 96.5$\pm$0.7 | 96.6$\pm$0.9 |
> | Random color 1 | 94.5$\pm$1.3 | 95.8$\pm$0.8 | 95.0$\pm$1.2 | 95.3$\pm$1.0 |
> | Random color 2 | 95.2$\pm$1.5 | 96.5$\pm$1.1 | 95.2$\pm$1.4 | 95.8$\pm$1.3 |
> | Random color 3 | 95.4$\pm$1.7 | 96.6$\pm$0.9 | 95.9$\pm$1.5 | 95.9$\pm$1.1 |
> | Random order 1 | 95.5$\pm$1.0 | 96.8$\pm$0.8 | 95.7$\pm$1.0 | 96.2$\pm$0.9 |
> | Random order 2 | 95.2$\pm$0.7 | 96.6$\pm$0.6 | 95.6$\pm$0.8 | 96.0$\pm$0.7 |
> | Random order 3 | 95.4$\pm$0.8 | 96.6$\pm$0.7 | 95.8$\pm$1.1 | 96.1$\pm$0.7 |
>
> **Line width/style and marker size/style:**
> We configured line style to solid, and marker as '*'. For the P12 and P19 datasets, the line width and the marker size are set to 1 and 2, while for the PAM dataset, they are set to 0.5 and 0.5. The guiding principle is to optimize the line graphs for human visualization, ensuring that the lines effectively show trends and markers clearly indicate observed data without obscuring the lines. We also tested the other configuration. Due to limited space, we only show the results on P19 here.
>
> | Line style | Line width | Marker | Marker size | AUROC | AUPRC |
> | --- | --- | --- | --- | --- | --- |
> | Solid | 1 | * | 2 | 89.4$\pm$1.9 | 52.8$\pm$3.8 |
> | Dotted | 1 | * | 2 | 88.7$\pm$2.0 | 53.3$\pm$3.4 |
> | Dashed | 1 | * | 2 | 88.5$\pm$2.8 | 53.3$\pm$4.2 |
> | Solid | 2 | * | 2 | 88.5$\pm$2.3 | 53.6$\pm$3.4 |
> | Solid | 3 | * | 2 | 88.9$\pm$1.9 | 52.8$\pm$3.2 |
> | Solid | 1 | o | 2 | 88.8$\pm$2.6 | 53.2$\pm$3.8 |
> | Solid | 1 | ^ | 2 | 88.8$\pm$2.2 | 53.1$\pm$4.0 |
> | Solid | 1 | * | 1 | 88.6$\pm$2.4 | 52.6$\pm$3.6 |
> | Solid | 1 | * | 3 | 89.2$\pm$2.1 | 53.1$\pm$4.2 |
>
> From the result, we can see that the line style has a significant impact on the performance, while markers do not. As for other factors, a good combination of line width and marker size might lead to better performance. Our approach can achieve competitive results in most settings.
>
> **Response to W2: more precise approach for image rendering.**
>
> We greatly value your insightful feedback. Our current study is a pivotal first step in investigating the potential of vision transformers applied to images transformed from time series. Currently, we've adopted a straightforward approach using matplotlib for image rendering, and our results suggest the feasibility and potential of this method. In light of your suggestion, we are considering directly assigning pixel values for image rendering, mirroring prior work on synthetic images used in vision model pre-training [1]. We leave it as future work.
>
> **Response to Q1: How were line colours selected?**
>
> As highlighted in our response to W1, for datasets with a limited number of variables, we manually selected the colors, ensuring that adjacent line graphs had distinguishable colors. For datasets with a larger number of variables, we divided the RGB value range into different slices and shuffled them, ensuring neighboring time series have distinct RGB values.
>
> **Response to Q2: For your self-supervised learning experiments, did you start with a pretrained model?**
>
> Yes, we start with the pre-trained checkpoint. It is an exploration of the potential of integrating vision techniques into the time series domain. Similar to masked image modeling work [2] in CV, we start with the pre-trained checkpoint and apply self-supervised learning.
>
> **References**
>
> [1] Kataoka, Hirokatsu, et al. "Pre-training without natural images." Proceedings of the Asian Conference on Computer Vision. 2020.
>
> [2] Xie, Zhenda, et al. "Simmim: A simple framework for masked image modeling." Proceedings of the IEEE/CVF Conference on Computer Vision and Pattern Recognition. 2022.

---

> > ### Comment · Reviewer_GiK3 · 2023-08-15
> >
> > Thank you for addressing my questions. The additional ablations look to be in line with the existing ones in the paper - it's good to have a more detailed understanding of the effects of these design choices, but some of them still show a somewhat concerning degree of variance. It makes sense that clarity to humans might just be the best heuristic here.
> >
> > My overall assessment is still the same: the general findings of the paper are original and interesting, and the presentation and reproducibility are very good. I still have some practical concerns around robustness and I think limitations should have been discussed.

---

### Official Review · Reviewer_QWW4 · 2023-07-25

**Soundness:** 3 good
**Presentation:** 3 good
**Contribution:** 3 good
**Rating:** 8
**Confidence:** 5

**Summary:**

For the classification task based on irregular time series data, the authors introduced Vision Time Series Transformer (ViTST) approach where irregular time series data is displayed as line graph then fed to pretrained transformer type models. The authors tested this approach using several time series data from medical domain and human activity domain, and the result shows confident advantages of the proposed approach.

**Strengths:**

Simple idea but very interesting. Could contribute to many domains dealing with time series data.

**Weaknesses:**

Time series data is everywhere but the authors tested only a few dataset from medical domain and human activity domain. Thus "any shape" in line 357 sounds too strong at this time.

**Questions:**

Have you tested multi-class classification tasks and/or regression tasks?

**Limitations:**

Since this approach could have a potential to expand widely, it would be better to state "Limitation" of this paper.

---

> ### Author Rebuttal · Authors · 2023-08-09
>
> Thank you for valuing our work and recognizing its strengths. We aim to address your concerns with our response below.
>
> **Response to W1: Time series data is everywhere but the authors tested only a few dataset from medical domain and human activity domain. Thus "any shape" in line 357 sounds too strong at this time.**
>
> In addition to the three irregularly sampled datasets P19, P12, and PAM, we also evaluated our approach on ten representative multivariate time series datasets from the UEA Time Series Classification Archive [1], as outlined in **Section 4.4**. These datasets have diverse characteristic in terms of number of variables, length, training size, and number of classes. Table 4 shows both the number of variables, length of different datasets and the performance comrasion on these datasets. More details on the datasets can be found in the Table 7 in the Appendix. For clarity, we also list it here:
> | Dataset | #Variables | #Classes | Length | Train Size |
> | --- | --- | --- | --- | --- |
> | EC | 3 | 4 | 1,751 | 261 |
> | UW | 3 | 8 | 315 | 120 |
> | SCP1 | 6 | 2 | 896 | 268 |
> | SCP2 | 7 | 2 | 1,152 | 200 |
> | JV | 12 | 9 | 29 | 270 |
> | SAD | 13 | 10 | 93 | 6599 |
> | HB | 61 | 2 | 405 | 204 |
> | FD | 144 | 2 | 62 | 5890 |
> | PS | **963** | 7 | 144 | 267 |
> | EW | 6 | 5 | **17984** | 128 |
>
> Table 4 in our paper presents the performance comparison of ViTST against six baseline methods designed for regular time series classification. From the table, we can see that our approach achieves consistently strong performance. Its average performance over the ten datasets are second-best, only slightly lower than TST. Notably, PS has the highest number of variables at 963, while EW boasts a longest sequence length of 17,984. Our approach still achieves competitive results on both, suggesting its effectiveness in handling time series with massive variables and/or long sequence.
> In summary, we assessed our technique on three datasets for irregularly sampled classification and ten for regular time series, covering a range of ''shapes''. Our approach consistently achieve competitive results. It's important to emphasize that most methods tailored for irregularly sampled time series often struggle with regularly sampled data, and vice versa.
>
> **Response to Q1: Have you tested multi-class classification tasks and/or regression tasks?**
>
> Yes, we tested multi-class classification tasks on multiple datasets, as shown in table above. EC, UW, JV, SAD, PS, EW, and PAM datatsets are all used for multi-class classification.
>
> **References**
>
> [1] Bagnall, Anthony, et al. "The UEA multivariate time series classification archive, 2018." arXiv preprint arXiv:1811.00075 (2018).

---

> > ### Comment · Reviewer_QWW4 · 2023-08-17
> >
> > Thanks for the responses. As the other reviewers also pointed, to state the limitation in the main text would be quite important since this paper will be a good starting point for the field. I'd like to keep my rating.

---

### Official Review · Reviewer_7PQE · 2023-07-26

**Soundness:** 2 fair
**Presentation:** 3 good
**Contribution:** 2 fair
**Rating:** 6
**Confidence:** 3

**Summary:**

The authors propose a method to model irregularly sampled time series. The method is based on transforming numerical time series data to line graphs and then applying pretrained vision transformers to that data. For multivariate datasets, every variable is plotted separately, and plots are aggregated in a grid. The authors provide exhaustive evaluation on different datasets, including leaving-sensors-out settings and regularly sampled time series data.

**Strengths:**

The paper is well written and easy to understand. The method proposed by the authors is extremely simple, yet surprisingly effective on many time series modeling tasks, as shown by exhaustive experimental evaluation.

**Weaknesses:**

-	I don’t see the authors claims made in the abstract and introduction backed by their empirical results. It is difficult to judge, whether their proposed method indeed outperforms previous approaches “significantly” (L. 10, L. 43, L. 67) as such an analysis is missing. On most of the metrics for the P19 and P12 data at least, improvements do not look significant to me. On the PAM dataset results are likely significant (analysis is also missing here).
-	The PAM dataset (the only one where we likely actually see a significant improvement over Raindrop) is also the one with the fewest number of variables (17 vs 34 & 36 for P19 and P12, respectively). I assume that the proposed method performs much worse for datasets with many variables, as also shown in the authors experiments on the FD, and PS datasets, where ViTST performs indeed much worse than other methods.
-	The authors do not comment on the computational cost of their method, which is likely much higher than for most non-vision based approaches.
-	Recently, Semenoglou et al. proposed a very similar method, albeit limited to single-variable time series. They plot a univariate time series and feed this plot into a CNN to predict future observations of the same series. While I think that the present work improves on the work by Semenoglou et al. in several regards (multivariate, pretrained ViT outperforming pretrained ResNet, …), the authors should include a reference to Semenoglou et al. in their related work. Additionally, I don't think statements such as "This paper studies the problem from a whole new perspective by transforming irregularly sampled time series into line graph images and leveraging powerful vision transformers for time series classification in the same way as image classification." (L. 4-7) are valid, given that the idea to transform time series to images and then applying image classification networks is not novel.
-	The authors do not comment on limitations of their method.

**Questions:**

-	For the PS dataset, the authors must have used much smaller (I assume 12x12) sizes for their plots, am I correct?
-	What is the computational cost of the proposed method?
-	Did the authors conduct experiments on whether the order of the plots in the grid matters?
-	Based on the ablation studies provided by the authors it seems likely that most of the performance comes from pretraining of the transformer models. Wouldn’t a large pretrained transformer for numerical time series modeling be likely to perform similar or better than the method proposed by the authors? If the authors agree, then I would like to see a brief discussion of this in an updated version of the manuscript.

**Limitations:**

Unfortunately, the authors do not comment on limitations of their work. I see the following possible limitations and would like to see an open discussion of those in an updated version of the manuscript:
-	Ability to handle extremely large number of variables
-	Computational costs of the method

---

> ### Author Rebuttal · Authors · 2023-08-09
>
> We sincerely appreciate your valuable feedback. Below are our responses to your concerns:
>
> **Response to W1: Concerns on “significant” improvement**
>
> We base our claim of "significant" improvement on extensive comparisons across three datasets and their associated metrics. In all the evaluated datasets and associated metrics, our method consistently shows superior performance.
> - P19 dataset: Our method outperforms the second-best approach, Raindrop, by 2.4 points in AUROC and 1.0 points in AUPRC.  In addition, our approach has a lower variance than Raindrop and the third-best approach DGM2-O.
> - PAM dataset: The advantage of our approach over the compared baselines are especially signification.
> - P12 dataset: While Raindrop is considered a strong baseline on P19 and PAM datasets, it lags behind our method by 2.8 points in AUROC and 5.8 in AUPRC. DGM2-O and mTAND are respectively the second-best in AUROC and AUPRC for this dataset. However, neither of the baselines achieves reasonable performance on both AUROC and AUPRC metrics.
>
> In conclusion, none of the specialized methods we evaluated consistently matched or surpassed our method across all metrics and datasets. Beyond the numbers, what we want to show is that our simple approach can match or surpass specialized methods regardless of the margin.
>
> **Response to W2: Performance on datasets with many variables**
>
> While the number of variables may play a role, other factors could also influence why our method's advantage on the P12 and P19 datasets isn't as significant as on the PAM dataset, such as the number of observations and/or the length of the time series.
> Regarding the PS dataset with the largest number of variables (963), our approach achieves better performance than the strong baseline TST and is second-best only behind XGBoost. On the FD dataset, it is comparable to XGBoost and ROCKET. It's worth noting that different models may be optimized for specific data types. For instance, our method excels in handling long sequences (EW dataset with a sequence length of 17984). When evaluating performance across all datasets, our method is second-best.  It's important to emphasize that most methods tailored for irregularly sampled time series often struggle with regularly sampled data, and vice versa.
>
> **Response to W3: Computational cost**
>
> Our experiments were conducted on Nvidia A6000 GPUs. It takes around 24 mins to run 20 epochs on the PAM dataset with a batch size of 72,  32 mins to run 4 epochs on the P12 dataset, and 58 mins to run 2 epochs (with upsampling) on the P19 dataset—all on a single GPU. While our computational costs might be higher than some non-vision methods, they appear to be within the range that could be considered acceptable in the context of current ML practices, considering the widespread use of large language and vision models nowadays.
>
> **Response to W4/5: Related work and limitations**
>
> We will add the work of Semenoglou et al. in the related work and add the limitation section in the updated version.
>
> **Response to Q1: For the PS dataset, the authors must have used much smaller (I assume 12x12) sizes for their plots, am I correct?**
>
> One can choose any size for the plot. For example, we tried the size of each plot (grid cell) as 24x24, where the image size increased to 768x768. With this adjustment, the performance of ViTST improves from 91.3 to 92.4, compared with using a 12x12 plot size.
>
> **Response to Q2: Computation cost**
>
> Please refer to the response to W3.
>
> **Response to Q3: Impact of grid order on results**
>
> We have conducted experiments to examine the impact of the grid order. By default, the grid order was sorted based on the number of observations. We also conducted tests with grids in random orders. The results from these tests are outlined below:
> | **P19** | **AUROC** | **AUPRC** |
> | --- | --- | --- |
> | Sorted order | 89.4$\pm$1.9 | 52.8$\pm$3.8 |
> | Random order 1 | 88.3$\pm$1.8 | 49.9$\pm$3.2 |
> | Random order 2 | 88.2$\pm$1.6 | 51.0$\pm$3.9 |
>
> | **P12** | **AUROC** | **AUPRC** |
> | --- | --- | --- |
> | Sorted order | 85.6$\pm$1.1 | 49.8$\pm$2.5 |
> | Random order 1 | 84.3$\pm$2.2 | 48.0$\pm$4.5 |
> | Random order 2 | 84.2$\pm$1.7 | 47.6$\pm$3.7 |
>
> | **PAM** | **Accuracy** | **Precision** | **Recall** | **F1 Score** |
> | --- | --- | --- | --- | --- |
> | Default order | 96.1$\pm$0.7 | 96.8$\pm$1.1 | 96.5$\pm$0.7 | 96.6$\pm$0.9 |
> | Random order 1 | 95.5$\pm$1.0 | 96.8$\pm$0.8 | 95.7$\pm$1.0 | 96.2$\pm$0.9 |
> | Random order 2 | 95.2$\pm$0.7 | 96.6$\pm$0.6 | 95.6$\pm$0.8 | 96.0$\pm$0.7 |
>
> The order does affect performance, most notably in the P19 and P12 datasets, which have varying observations across different variables. On the other hand, the PAM dataset, with a uniform observation count for every variable, displays less variability in the performance. A possible explanation for this could be that clustering regions with higher observation plots, which are also more informative, allows the model to better capture their correlations and allocate attention effectively.
>
> **Response to Q3: Effectiveness of a large pretrained transformer for numerical time series modeling**
>
> A large pre-trained transformer for numerical time series modeling might offer comparable or superior performance to ViTST. However, such a transformer would necessitate extensive time series data for pretraining, requiring significant data collection efforts. It's important to emphasize that our intention is not to advocate that our vision transformer-based approach is the optimal option for performance. Instead, we aim to demonstrate that the publicly available pre-trained vision transformer can easily adapt to time series modeling and achieve satisfying results. This method largely simplifies model design and data collection efforts and could also bridge the time series and computer vision domain, opening up the possibility of adapting established vision techniques to the time series domain and facilitating multi-modal research.

---

> > ### Comment · Reviewer_7PQE · 2023-08-16
> >
> > I would like to thank the authors for their detailed response and the additional experiments. I find the results on the impact of grid order quite interesting and think the authors explanation for this is plausible.
> >
> > Can the authors also comment on the computation cost of their method for inference compared to other non-vision based methods?
> >
> > I fully agree with the authors response to my Q3. Please include this important point in the discussion/limitation section!

---

> > > ### Author Response · Authors · 2023-08-18
> > > **Response to the reviewer's reply**
> > >
> > > Thank you for your thoughtful feedback and for taking the time to review the additional experiments. We appreciate your agreement with our explanations. We will ensure that a discussion on the limitations of the current approach is incorporated in the updated version of the paper.
> > >
> > > As for the inference cost comparison, we list the inference time (in seconds) of different methods on the test sets of three datasets below. All the inferences are made in a single Nvidia A6000 GPU.
> > >
> > > | Datatset | Transformer | mTAND | SeFT | Raindrop | MTGNN | DGM^2-O | GRU-D | ViTST |
> > > | --- | --- | --- | --- | --- | --- | --- | --- | --- |
> > > | P19 | 0.21 | 0.52 | 2.72 |  3.05 | 3.62 |  2.47 | 31.04 | 44.51 |
> > > | P12 | 0.12 |  0.44 | 0.97 | 1.27 | 1.46 | 2.80 | 10.13 | 12.14 |
> > > | PAM | 0.06 | 0.23 | 0.89 | 0.67 | 1.16 | 2.98 | 4.55 | 5.30 |
> > >
> > > Our vision-based method consumes more inference time than the non-vision baselines. However, we believe this cost remains within an acceptable range in the context of today's ML practice and medical applications. We will include the computation cost in the updated version of the paper.
> > >
> > > Once again, thank you for your valuable insights.

---

> > > > ### Comment · Reviewer_7PQE · 2023-08-19
> > > >
> > > > I would like to thank the authors for their reply and reporting their inference times. As expected their method is up to 2 orders of magnitudes slower than comparison baselines. Although I agree with the authors that this may not be a serious practical issue, I think it is extremely important to include an **open and honest** discussion about this and other limitations of their work in a separate section in the paper. Trusting that such a discussion will be included, I'll increase my score to 6 since the authors addressed most of my concerns successfully.

---

### Decision · Program_Chairs · 2023-09-21

**Decision:**

Accept (poster)

**Comment:**

The idea behind this paper is simple. When making predictions with time series data, instead of using a time series model, use matplotlib to plot the time series on an image and then use a ViT to predict outcomes. The paper studies variations on this problem including the effect of pretraining the vision transformers on Imagenet, comparisons against several time series models and datasets (in healthcare and outside of it) showcase that the model can capture signal from time series data. Most reviewers liked the work, in particular its simplicity and broad applicability.

During the discussion phase, the reviewers asked for robustness results obtained by varying plotting artefacts which the rebuttal provided, reviewers also requested predictive models on MIMIC which was not provided in the rebuttal.
The claims in the paper were requested to be toned down given the paper does not present a rigorous study to quantify the kinds of patterns it works well/less well on.
Given the simplicity of the proposal, two reviewers independently ran the code to check if it works and it reproduces the reported results. After some back and forth, the reviewers verified that the work does run (using the authors' instructions) and produce useful results but highlighted that the codebase is not well organized and the results are not SOTA but rather in the same ballpark as existing prior work.
More importantly, the reviewers found that the model was setup to work using statistics obtained on the train and test set which violates the typical practice of normalization using training set statistics. This is an oversight and should be fixed in all the experiments (the rebuttal focused on P19) since it could lead to artificial inflation of results and unfairly disadvantages other baselines. This is also a practical concern about the utility of the model. There are scenarios where test patients' biomarkers are significantly above or below what happens to patients at training time. When this happens, without using test statistics for normalization (which is incorrect from a modeling perspective), there will invariably time series data at test time that goes outside the image (out of bounds) and remain invisible to the model.
At the end of the discussion period, this remains a truly borderline paper. the reviewers remained split along whether the paper should be accepted or not where one can argue for acceptance based on the simplicity of ideas or ask for a revamp based on cleaning up the code further.

On the whole I believe this is a very strong and simple idea which will have an impact even if it is imperfect.

The manuscript should be updated (in terms of toning down the claims made) and incorporated with changes made to the experimental setup to correct for the normalization issue (which seems like it might affect all the experimental results). Attention should be paid to the code so that it is accessible enough to reliably reproduce the reported results.